# Meta-Learning Approach for Joint Multimodal Signals with Multimodal Iterative Adaptation

**Sehun Lee**[*]                                                                          *yhytoto12@snu.ac.kr*
*Department of Computer Science and Engineering*
*Seoul National University*

**Wonkwang Lee**[*]                                                                    *wonkwang.lee@snu.ac.kr*
*Department of Computer Science and Engineering*
*Seoul National University*

**Gunhee Kim**                                                                          *gunhee@snu.ac.kr*
*Department of Computer Science and Engineering*
*Seoul National University*

**Reviewed on OpenReview:** *https://openreview.net/forum?id=LVO4KBaIQt*

## Abstract

In the pursuit of effectively modeling real-world joint multimodal signals, learning to learn multiple Implicit Neural Representations (INRs) jointly has gained attention to overcome data scarcity and enhance fitting speed. However, predominant methods based on multimodal encoders often underperform due to their reliance on direct data-to-parameter mapping functions, bypassing the optimization steps necessary for capturing the complexities of real-world signals. To address this gap, we propose Multimodal Iterative Adaptation (MIA), a novel framework that combines the strengths of multimodal fusion with optimization-based meta-learning. The key idea is to enhance the learning of INRs by facilitating exchange of cross-modal knowledge among learners during the iterative optimization processes, improving generalization and enabling a more nuanced adaptation to complex signals. To achieve this, we introduce State Fusion Transformers (SFTs), an attention-based meta-learner designed to operate in the backward pass of the learners, aggregating learning states, capturing cross-modal relationships, and predicting enhanced parameter updates for the learners. Our extensive evaluation in various real-world multimodal signal regression setups shows that MIA outperforms existing baselines in both generalization and memorization performances. Our code is available at https://github.com/yhytoto12/MIA.

## 1 Introduction

Implicit neural representations (INRs) are a class of neural networks designed to represent signals or data as continuous coordinate-to-feature mapping functions (*e.g.* an image is a function $I(x, y) = (r, g, b)$ mapping 2D coordinates to color intensity values). INRs offer several advantages over traditional data representations (*e.g.* discrete 2D arrays for images), including the ability to recover inherently continuous signals that are often sampled sparsely and stored discretely (*e.g.* videos are the recordings of dynamic scenes that change continuously over space and time) and better scaling property with signal resolution (Dupont et al., 2022a).

Since a variety of real-world signals can be represented as such continuous functions or they even inherently arise from such functions, numerous efforts have been made to formulate data as functions and represent them with INRs in a wide variety of domains or modalities, including audios (Sitzmann et al., 2020), time series (Fons et al., 2022), images (Sitzmann et al., 2020), videos (Chen et al., 2021a), 3D geometries (Park et al.,

---

[*]Equal Contribution

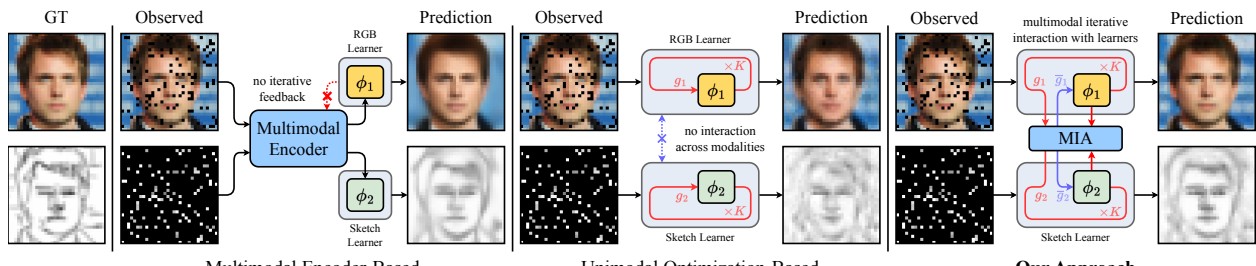

Figure 1: Left: Multimodal encoder-based methods predict INR parameters directly from signals, omitting optimization steps for rapid adaptation, often failing to fully capture complexities of real-world signals (*e.g.* blurry RGB prediction). Middle: Unimodal methods struggle with generalization in data-sparse scenarios due to their lack of cross-modal interactions (*e.g.* inaccurate Sketch prediction). Right: Our Multimodal Iterative Adaptation (MIA) enhances learner interaction and cross-modal knowledge exchange during the backward optimization pass, leading to improved fitting and generalization.

2019), 3D scenes (Mildenhall et al., 2020), and 3D motions (Pumarola et al., 2021). They have shown promise in various practical applications that require the precise modeling of signals and their inherently continuous properties, such as generation (Yu et al., 2022), compression (Kim et al., 2022b), super-resolution (Chen et al., 2021b), video super-slomo (Chen et al., 2022b), and novel-view synthesis (Takikawa et al., 2021).

As the field of INRs advance, learning to learn a group of multiple INRs jointly has gained attention recently to enhance convergence and data efficiency when modeling joint multimodal signals (Kim et al., 2022a; Shen et al., 2023). Such signals, often arising from correlated multimodal distributions with interdependent information, are crucial in diverse areas such as embodied agents (Xia et al., 2018), climate analysis (Wang et al., 2019), healthcare (Muhammad et al., 2021), neuroscience (Ulrich et al., 2015), drug discovery (Chan et al., 2023), and smart-grid systems (Kamarthi et al., 2022).

In modeling such joint multimodal signals, the primary approach involves encoder-based meta-learning methods (Kim et al., 2022a; Shen et al., 2023) that use multimodal encoders to predict INR parameters directly from data through attention-based fusion. Concurrently, optimization-based methods (Tancik et al., 2021; Dupont et al., 2022a), though typically unimodal, focus on learning an effective parameter initialization to speed up convergence and enhance generalization. These methods iteratively optimize INR parameters from this initialization using gradient descent algorithms at test time.

Nevertheless, both approaches come with their own limitations, as illustrated in Figure 1. Encoder methods primarily rely on direct data-to-parameter mappings without optimization steps for rapid adaptation, often failing to capture complex signals (e.g., underfitting high-frequency details) (Kim et al., 2019). In contrast, unimodal methods, despite their rapid signal fitting capability thanks to iterative optimization processes from good initializations, lack mechanisms for leveraging interacting multimodal structures in data, limiting their generalization in data-scarce situations (Kim et al., 2022a).

To bridge these gaps, we present a novel optimization-based meta-learning framework called Multimodal Iterative Adaptation (MIA). Unlike the encoder-based methods, MIA builds upon the advantages of optimization-based meta-learning methods, *i.e.* meta-learning good initializations and adapting them to signals through iterative optimization steps for accurate fitting. In contrast to the unimodal counterparts, however, MIA further empowers each modality learner to interact and inform one another during adaptation. This fosters a rich exchange of information and synergistic learning of INRs across different modalities, leading to enhanced generalization capability with limited observations.

The core of MIA is an attention-based meta-learner coined as State Fusion Transformers (SFTs), particularly designed to operate in the backward pass of the INR learners (*i.e.* within the space of learners' parameters and gradients). SFTs are meta-learned to achieve three goals: (1) aggregating the current *learning states* (*i.e.* parameters and gradients) of the learners, (2) capturing modality-specific and cross-modal relationships between those states through attention mechanisms, and (3) utilizing this knowledge to predict better parameter updates for each learner, facilitating cross-modal interactions at each iterative adaptation step.

It's noteworthy to mention that attention mechanisms are well-studied across diverse multimodal learning contexts and applications. However, their potential to uncover and utilize interdependencies within multiple optimization problems and their trajectories (often represented by *states* of the learners for each optimization step) remains relatively unexplored. Our work contributes uniquely to the literature by being the first to investigate the integration of those potential multimodal structures within the scope of jointly learning multiple correlated INRs, showing how optimizing for one modality can inform and enhance solutions for others. While doing so, we detail recipes for effectively identifying and leveraging these multimodal relationships, including techniques for scaling gradients properly for each modality and integrating original, unimodal, and multimodal learner states. We believe these insights offer interesting new directions in the study of interacting optimization problems, especially useful when data scarcity could lead to bad optimization solutions.

In experiments, we apply our technique to prior unimodal INR meta-learning frameworks such as Functa and Composers, and evaluate them on a variety of multimodal signal regression scenarios, including modeling 1D synthetic functions, ERA5 global climate data, 2D CelebA visual images, and Audiovisual-MNIST data. The results consistently demonstrate that our MIA significantly enhances the learning capabilities of the unimodal baselines. Moreover, it outperforms the encoder-based baselines in terms of both generalization and memorization performance. We also conduct in-depth and wide-ranging analysis studies to validate the necessity of each component in SFTs for achieving these improvements.

## 2 Related Work

### 2.1 Meta-learning for INRs.

Fitting INRs to a function from randomly initialized weights has been found to be notoriously inefficient; it requires hundreds or thousands of optimization steps to converge (Sitzmann et al., 2020; Mildenhall et al., 2020), and learned representations do not generalize well on novel input coordinates under the few-shot learning regimes (Jain et al., 2021). To overcome such limitations, there have been increasing studies to apply meta-learning to facilitate the learning of INRs. The methods typically include either (1) meta-learning an encoder to infer the parameters of INRs directly from observed data (Chen & Wang, 2022; Kim et al., 2023) or optimization-based approach where a weight initialization of INRs is learned to enable rapid adaptation to signals with a few optimization steps (Tancik et al., 2021; Bauer et al., 2023).

### 2.2 Multimodal Learning.

Multimodal learning aims to integrate data or representations from multiple modalities to enhance the robustness and accuracy of learning systems, leveraging the complementary information available across different data types. Most of the work in the literature typically focus on learning multimodal encoders to infer enhanced representations from observed data, where attention mechanisms are utilized for multimodal fusion (Liang et al., 2022; Bachmann et al., 2022). Unlike this traditional focus on enhancing representations of models in their forward pass, we explore a novel setup where we utilize attention modules to enhance the learning of the models in their backward pass at each iterative optimization step.

### 2.3 Learning to Optimize (L2O).

Our method also aligns with the works in L2O, another meta-learning domain that focuses on enhancing an optimization algorithm rather than weight initialization. Its idea is to empower an optimizer with neural networks that meta-learn useful prior knowledge on the optimization process (Andrychowicz et al., 2016). To achieve that idea, Metz et al. (2022a) utilize various state features of learners, including parameters, gradients and their higher-order statistics. In addition, Kang et al. (2023) meta-learn a gradient preconditioning method for improved convergence and Baik et al. (2023) introduce a meta-learned mechanism to estimate weight decaying and learning rate factors for improved generalization. Distinct from these methods, our work takes an intesting orthogonal direction by exploring the potential of leveraging inter-modal interactions among state features of learners to address data scarcity issues.

## 3 Preliminaries

### 3.1 Implicit neural representations (INRs)

INRs are a class of neural networks, often parameterized as a stack of MLPs, that are designed to approximate a function $f : x \mapsto y$ mapping the coordinates $x$ to the features $y$. We interchangeably use data, signals, and functions throughout the paper. Given a set of $P$ coordinate-feature pairs $\mathcal{D} = \{(x^i, y^i)\}_{i=1}^{P}$ of a function, an INR $f_{\text{INR}}(\cdot; \theta)$ parameterized by $\theta$ is optimized to minimize a loss function as below:

$$\mathcal{L}(\mathcal{D}; \theta) = \frac{1}{P} \sum_{i=1}^{P} ||f_{\text{INR}}(x^i; \theta) - y^i||_2^2. \tag{1}$$

Despite the recent advances, optimizing individual INRs from randomly initialized weights are notoriously inefficient (Sitzmann et al., 2020; Mildenhall et al., 2020). Also, they often do not generalize well when observations are sparse (Tancik et al., 2021; Jain et al., 2021), *i.e.* when $P$ is small. To address these problems, optimization-based meta-learning techniques have been proposed.

### 3.2 Meta-Learning Approach

Two notable examples are Functa (Dupont et al., 2022b;a; Bauer et al., 2023) and Composers (Kim et al., 2023). These methods build upon CAVIA (Zintgraf et al., 2018) and implement two key ideas: (1) Meta-learning INR parameters $\theta$ that capture data-agnostic priors, enabling faster adaptation when modeling each signal. (2) Introducing additional context parameters $\phi \in \mathbb{R}^{S \times D}$, a set of $D$-dimensional features that are adapted to each signal. These features encapsulate data-specific variations and conditions of the meta-learned INRs $f_{\text{INR}}(\cdot; \theta)$ via modulation schemes (Perez et al., 2018; Schwarz et al., 2023) during the adaptation stage.

Formally, given a dataset $\mathcal{D} = \{\mathcal{D}_n\}_{n=1}^{N}$ of $N$ functions, the objective in Eq. 1 is extended to accommodate both $\theta$ and $\phi_n$ for each function $\mathcal{D}_n$ as follows:

$$\mathcal{L}(\mathcal{D}_n; \theta, \phi_n) = \frac{1}{P_n} \sum_{i=1}^{P_n} ||f_{\text{INR}}(x_n^i; \theta, \phi_n) - y_n^i||_2^2. \tag{2}$$

Then, INR weights $\theta$ and initial context parameters $\phi$ are optimized with the meta-objective as below:

$$\theta, \phi = \underset{\theta, \phi}{\operatorname{argmin}} \, \mathbb{E}_n \left[ \mathcal{L}(\mathcal{D}_n; \theta, \phi_n^{(K)}) \right], \text{ where} \tag{3}$$

$$\phi_n^{(k+1)} = \phi_n^{(k)} - \alpha \cdot \nabla_{\phi_n^{(k)}} \mathcal{L}(\mathcal{D}_n; \theta, \phi_n^{(k)}), \tag{4}$$

for $k = 0, \ldots, K - 1$. $\alpha$ is a learning rate and $\phi_n^{(0)} = \phi$. Intuitively, the objective can be cast as a bi-level optimization problem: (1) In the inner loop (Eq. 4), each INR learner is initialized with weights $\theta$ and $\phi_n^{(0)}$ and then adapted to each function $\mathcal{D}_n$ through $\phi_n^{(k)}$ ($k = 1, \ldots, K$). (2) In the outer loop (Eq. 3), the meta learner updates the INR weights $\theta$ and initial context parameters $\phi$ so that each fitted INR $f_{\text{INR}}(\cdot; \theta, \phi_n^{(K)})$ recovers the signal $\mathcal{D}_n$ well within $K$ steps.

## 4 Approach

We delve into our optimization-based meta-learning framework for learning multiple INRs jointly when modeling multimodal signals. We first extend previous meta-learning frameworks to multimodal setups. Then, we introduce our MIA, its core components SFTs, and the meta-learning objectives. We present a schematic illustration in Figure 2 and provide a meta-learning algorithm of our MIA in Algorithm 1.

We consider a dataset $\mathcal{D} = \{\mathcal{D}_n\}_{n=1}^{N}$ with $N$ pairs of multimodal signals. Each pair $\mathcal{D}_n = \{\mathcal{D}_{nm}\}_{m=1}^{M}$ is composed of signals from $M$ different modalities. Each signal $\mathcal{D}_{nm} = \{(x_{nm}^i, y_{nm}^i)\}_{i=1}^{P_{nm}}$ contains $P_{nm}$ coordinate-feature pairs which could vary with signals.

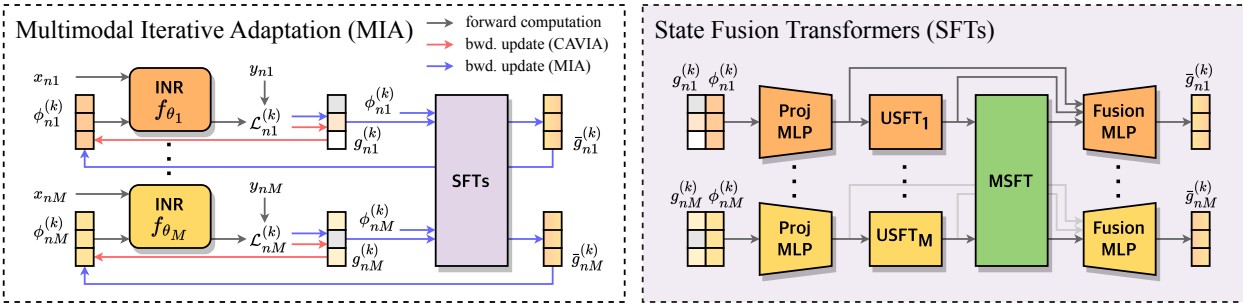

Figure 2: An illustration on our proposed Multimodal Iterative Adaptation (MIA). Left: At each step $k$, each INR learner ($f_{\theta_m}$) calculates the loss ($\mathcal{L}_{nm}^{(k)}$) given the weights ($\phi_{nm}^{(k)}$) and data ($x_{nm}, y_{nm}$) during the forward pass (black arrows), followed by computing gradients ($g_{nm}^{(k)}$) to update the weights. Before updating the weights with the gradients (red arrows), we enhance them via State Fusion Transformers (SFTs) (blue arrows). Right: SFTs aggregate the states ($\{\phi_{nm}^{(k)}, g_{nm}^{(k)}\}_m$), capture the unimodal and cross-modal dependencies via USFTs and MSFTs, and fuse this knowledge to compute enhanced parameter updates ($\{\bar{g}_{nm}^{(k)}\}_m$) via Fusion MLPs.

## 4.1 A Naive Framework for Multimodal Signals

We begin with a simple baseline that combines per-modality meta-learners together while treating each of them separately. This framework consists of a set of independent per-modality INR $f_{\text{INR}}(\cdot; \theta_m, \phi_m)$ with parameters $\theta = \{\theta_m\}_{m=1}^M$ and initial context parameters $\phi = \{\phi_m\}_{m=1}^M$. Then, we modify the meta-objective to promote both memorization and generalization performances during the adaptation. For this, we split each data into support $\mathcal{D}_{nm}^{\text{train}}$ and query $\mathcal{D}_{nm}^{\text{val}}$ sets, and use the following bi-level optimization algorithm to meta-learn the parameters for each modality:

$$\theta_m, \phi_m = \underset{\theta_m, \phi_m}{\operatorname{argmin}} \, \mathbb{E}_n \left[ \mathcal{L}(\mathcal{D}_{nm}^{\text{val}}; \theta_m, \phi_{nm}^{(K)}) \right], \text{ where} \tag{5}$$

$$\phi_{nm}^{(k+1)} = \phi_{nm}^{(k)} - \alpha_m \cdot g_{nm}^{(k)}, \quad g_{nm}^{(k)} = \nabla_{\phi_{nm}^{(k)}} \mathcal{L}(\mathcal{D}_{nm}^{\text{train}}; \theta_m, \phi_{nm}^{(k)}), \tag{6}$$

for $k = 0, \dots, K-1$ with $\phi_{nm}^{(0)} = \phi_m$. $\alpha_m$ is a learning rate for $\phi_{nm}^{(k)}$ of each modality.

## 4.2 Multimodal Iterative Adaptation (MIA)

We enable cross-modal interaction among the learners through our MIA for enhanced parameter updates. Our rationale is grounded by the fact that the parameters of one modality learner carry information on a comprehensive encoding of the signals within a modality, whereas the gradients highlight both the adequacy of this encoding via magnitude and the direction for improvement. Thus, by allowing access to the states from other modalities through our MIA, we enable each learner to benefit from a broader, richer context for its own updates. Essentially, this process encourages learners to not only optimize their individual parameters but also to contribute to and benefit from a collective improvement across modalities. This process is described as:

$$\bar{g}_{n1}^{(k)}, \dots, \bar{g}_{nM}^{(k)} = U_\xi \big( \{ (g_{nm}^{(k)}, \phi_{nm}^{(k)}) \}_{m=1}^M \big), \tag{7}$$

where $U_\xi$ denotes State Fusion Transformers (SFTs), a collection of meta-learned modules.

## 4.3 State Fusion Transformers (SFTs)

As shown in Figure 2, SFTs enhance the learning of INRs via a three-step process: (1) aggregating the states of the unimodal learners, (2) capturing modality-specific and cross-modal information within those states to promote the knowledge exchange across them via attention mechanisms, (3) and utilizing this knowledge

---

**Algorithm 1** Meta-Learning Algorithm of MIA.

---

**Require:** Batch size $B$; Inner step $K$; Outer learning rates $\lambda_\Theta$ ($\Theta = \{\theta, \phi, \xi, \eta\}$).
1: Initialize $\Theta = \{\theta, \phi, \xi, \eta\}$.
2: **repeat**
3:      Sample a batch of $B$ joint multimodal signals $\{\{\mathcal{D}_{nm}\}_{m=1}^M\}_{n=1}^B$.
4:      Sample a sampling ratio $R_{nm} \sim \mathcal{U}(R_m^{\min}, R_m^{\max})$ for $\forall n, m$.
5:      Split $\mathcal{D}_{nm}$ into $\mathcal{D}_{nm}^{\text{train}}, \mathcal{D}_{nm}^{\text{val}}$, where $|\mathcal{D}_{nm}^{\text{train}}| = P_{nm} \times R_{nm}$ and $|\mathcal{D}_{nm}^{\text{val}}| = P_{nm}$.
6:      **for** $n = 1, \dots, B$ **do**
7:          **for** $k = 0, \dots, K - 1$ **do**
8:              **for** $m = 1, \dots, M$ **do**
9:                  $g_{nm}^{(k)} = \nabla_{\phi_{nm}^{(k)}} \mathcal{L}(\mathcal{D}_{nm}^{\text{train}}; \theta_m, \phi_{nm}^{(k)})$                  ▷ Eq. 6
10:                 $g_{nm}^{(k)} \leftarrow \exp(\eta_m) \cdot g_{nm}^{(k)}$                      ▷ Eq. 12
11:                 $z_{nm}^{(k)} = \mathsf{ProjectMLP}_m(g_{nm}^{(k)}, \phi_{nm}^{(k)}; \xi_m)$
12:                 $\hat{z}_{nm}^{(k)} = \mathsf{USFT}_m(z_{nm}^{(k)}; \xi_m)$                       ▷ Eq. 8
13:              **end for**
14:              $\tilde{z}_{n1}^{(k)}, \dots, \tilde{z}_{nM}^{(k)} = \mathsf{MSFT}(\hat{z}_{n1}^{(k)}, \dots, \hat{z}_{nM}^{(k)}; \xi_s)$        ▷ Eq. 9
15:              **for** $m = 1, \dots, M$ **do**
16:                 $\bar{g}_{nm}^{(k)} = \mathsf{FusionMLP}_m([z_{nm}^{(k)} \| \hat{z}_{nm}^{(k)} \| \tilde{z}_{nm}^{(k)}]; \xi_m)$      ▷ Eq. 10
17:                 $\phi_{nm}^{(k+1)} \leftarrow \phi_{nm}^{(k)} - \bar{g}_{nm}^{(k)}$                   ▷ Eq. 11
18:              **end for**
19:          **end for**
20:      **end for**
21:      $\mathcal{L}_{\text{outer}} = \frac{1}{B} \sum_{n=1}^B \sum_{m=1}^M \mathcal{L}(\mathcal{D}_{nm}^{\text{val}}; \theta_m, \phi_{nm}^{(K)})$.            ▷ Eq. 11
22:      $\Theta \leftarrow \Theta - \lambda_\Theta \nabla_\Theta \mathcal{L}_{\text{outer}}$, where $\Theta = \{\theta, \phi, \xi, \eta\}$.        ▷ Eq. 11
23: **until** converged

---

to predict better parameter updates for the learners. SFTs consist of Unimodal State Fusion Transformers (USFTs), Multimodal State Fusion Transformers (MSFTs), and Fusion MLPs.

For each adaptation step $k$, we first compute per-modality state representations $z_{nm}^{(k)} \in \mathbb{R}^{S_m \times D_z}$ from the gradients and parameters of each modality learner. We concatenate the context parameters $\phi_{nm}^{(k)} \in \mathbb{R}^{S_m \times D_\phi}$ and gradients $g_{nm}^{(k)} \in \mathbb{R}^{S_m \times D_\phi}$ along the feature dimension, followed by a per-modality projection MLPs. Then, these state representations are fed into USFTs with per-modality $L_1$ transformer blocks ($\xi_m$):

$$\hat{z}_{nm}^{(k)} = \mathsf{USFT}_m(z_{nm}^{(k)}; \xi_m), \text{ for } m = 1, \dots, M, \tag{8}$$

where $\hat{z}_{nm}^{(k)}$ are updated state representations by USFTs, which model dependencies among a set of states within a modality. Next, we combine these per-modality representations into a sequence $\hat{z}_n^{(k)} = \{\hat{z}_{nm}^{(k)}\}_{m=1}^M$ and input to MSFTs with shared $L_2$ transformer blocks ($\xi_s$). These enhanced state representations $\tilde{z}_n^{(k)}$ further capture cross-modal relationships:

$$\tilde{z}_n^{(k)} = \mathsf{MSFT}(\hat{z}_n^{(k)}; \xi_s), \text{ where } \hat{z}_n^{(k)} = \{\hat{z}_{nm}^{(k)}\}_{m=1}^M. \tag{9}$$

The final step is to integrate them all and estimate enhanced parameter updates for the learners. We concatenate the computed per-modality state representations $z_{nm}^{(k)}, \hat{z}_{nm}^{(k)}, \tilde{z}_{nm}^{(k)} \in \mathbb{R}^{S \times D_z}$ along the feature dimension, followed by processing each feature with Fusion MLPs:

$$\bar{g}_{nm}^{(k)} = \mathsf{FusionMLP}(\bar{z}_{nm}^{(k)}; \xi_m), \text{ where } \bar{z}_{nm}^{(k)} = [z_{nm}^{(k)} \| \hat{z}_{nm}^{(k)} \| \tilde{z}_{nm}^{(k)}]. \tag{10}$$

Here, $\bar{g}_n^{(k)} = \{\bar{g}_{nm}^{(k)}\}_{m=1}^M$ is the predicted weight updates for the unimodal learners. They are used to adapt $\phi_{nm}^{(k)}$ for each inner step $k = 0, \dots, K - 1$, while the parameters of SFTs $\xi = \xi_s \cup \{\xi_m\}_{m=1}^M$ along with those

of INRs ($\theta$ and $\phi$) are meta-learned during outer-optimization:

$$\theta, \phi, \xi = \operatorname*{argmin}_{\theta,\phi,\xi} \mathbb{E}_{n,m} \left[ \mathcal{L}(\mathcal{D}_{nm}^{\text{val}}; \theta_m, \phi_{nm}^{(K)}) \right], \text{ where } \phi_{nm}^{(k+1)} = \phi_{nm}^{(k)} - \bar{g}_{nm}^{(k)}. \tag{11}$$

## 4.4 Gradient Scaling

The direct use of gradients as inputs to neural networks could be problematic due to the presence of extremely small values, leading to a bottleneck in learning process. Also, the gradient distributions could differ significantly across modalities, hindering their effective multimodal fusion. To remedy this, we meta-learn a scaling constant for each modality and rescale the gradients before feeding them into projection MLPs:

$$g_{nm}^{(k)} \leftarrow \exp\left(\eta_m\right) \cdot g_{nm}^{(k)}, \tag{12}$$

where we exponentiate the scaling constants $\{\eta_m\}_{m=1}^{M}$ to ensure positive rescaling and maintain gradient directions.

# 5 Experiments

We demonstrate our proposed MIA improves performance of state-of-the-art meta-learned INR models. We refer the reader to Appendix A for more qualitative samples.

**Baselines.** We adopt Functa (Dupont et al., 2022a) and Composers (Kim et al., 2023) as the base INR models, collectively referred to as CAVIA in subsequent sections. Then, we build our framework and other baselines upon this CAVIA. We include recent L2O methods such as MetaSGD (Li et al., 2017), GAP (Kang et al., 2023), and ALFA (Baik et al., 2023) as baselines integrated into CAVIA. These approaches aim to enhance learner's convergence and generalization by modulating gradients, yet they do not account for cross-modal interactions. We also compare against the state-of-the-arts encoder-based multimodal meta-learning models, HNPs (Shen et al., 2023) and MTNPs (Kim et al., 2022a), which are based on Neural Processes (NPs)(Garnelo et al., 2018). While HNPs are heavily demonstrated on multimodal classification, MTNPs are designed specifically for multimodal signal regression tasks like ours, using encoders with specialized attention and pooling instead of iterative optimization. We set $K = 3$ for optimization-based methods. See Appendix D.2 for detailed discussions on baselines.

**Datasets and Metrics.** Our method is evaluated across four datasets: (1) multimodal 1D synthetic functions (Kim et al., 2022a), (2) multimodal 2D CelebA images (Xia et al., 2021), (3) ERA5 global climate dataset (Hersbach et al., 2019), and (4) Audiovisual-MNIST (AV-MNIST) dataset (Vielzeuf et al., 2018). For training, we construct the support set $\mathcal{D}_{nm}^{\text{train}}$ by sampling $P_{nm} \times R_{nm}$ coordinate-feature pairs from each full dataset $\mathcal{D}_{nm}$. The sampling ratio $R_{nm}$ varies within predefined ranges $[R_m^{\min}, R_m^{\max}]$, independently drawn for each signal. The full dataset serves as the query set $\mathcal{D}_{nm}^{\text{val}}$ during validation to assess both memorization and generalization capabilities. We measure performance using mean squared errors (MSEs), averaged over 5 runs, across different ranges of sampling ratios. More details, including the number of signals ($N$), modalities ($M$), and coordinate-value pairs ($P$) for each dataset, can be found in Appendix D.1.

## 5.1 Multimoal 1D synthetic function regression

**Setups.** Following prior works (Finn et al., 2017; Guo et al., 2020; Kim et al., 2022a), we conduct experiments on a simple yet controlled 1D synthetic function regression setting. Specifically, we first define a canonical form of parametric multimodal functions as below:

$$y_{nm} = a_n \cdot \texttt{Act}_m(b_n \cdot x_{nm} + c_n) + d_n, \text{ where}$$
$$\texttt{Act}_m \in \{\texttt{Sine}, \texttt{Gaussian}, \texttt{Tanh}, \texttt{ReLU}\}. \tag{13}$$

Each function $\mathcal{D}_{nm}$ is instantiated with the shared parameters $(a_n, b_n, c_n, d_n)$ and modality-specific non-linear activation functions $\texttt{Act}_m(\cdot)$. We follow Kim et al. (2022a) to construct the dataset: We define a uniform grid of $P_{nm} = 200$ coordinates within a range of $x \in [-5, 5]$. We sample $N = 1000$ function parameters

Table 1: Quantitative comparisons on the multimodal 1D synthetic functions. We report the normalized MSEs ($\times 10^{-2}$) computed over distinct ranges of sampling ratios, averaged over 5 random seeds. The multimodal methods are marked with the dagger (†) symbol.

| Modality | | Sine | | | Gaussian | | | Tanh | | | ReLU | | |
|---|---|---|---|---|---|---|---|---|---|---|---|---|---|
| Range $R^{\min}$ $R^{\max}$ | | 0.01 0.02 | 0.02 0.05 | 0.05 0.10 | 0.01 0.02 | 0.02 0.05 | 0.05 0.10 | 0.01 0.02 | 0.02 0.05 | 0.05 0.10 | 0.01 0.02 | 0.02 0.05 | 0.05 0.10 |
| Functa | | 45.45 | 16.58 | 3.316 | 19.83 | 4.603 | 1.043 | 23.02 | 3.788 | 0.587 | 84.49 | 13.54 | 2.992 |
| w/ MetaSGD | | 44.69 | 16.46 | 3.326 | 19.75 | 4.334 | 1.055 | 26.16 | 4.528 | 0.718 | 69.86 | 10.45 | 2.044 |
| w/ GAP | | 44.76 | 16.49 | 3.297 | 19.54 | 4.518 | 1.052 | 23.46 | 3.799 | 0.555 | 63.17 | 10.12 | 1.862 |
| w/ ALFA | | 48.09 | 18.67 | 5.160 | 19.09 | 4.910 | 1.580 | 22.45 | 4.200 | 0.872 | 53.83 | 8.303 | 1.536 |
| w/ HNPs† | | 16.95 | 8.397 | 2.547 | 1.634 | 0.912 | 0.293 | 2.844 | 0.673 | **0.091** | 9.247 | **0.772** | **0.095** |
| w/ MTNPs† | | 13.34 | 4.718 | 1.871 | 2.019 | 1.513 | 1.285 | 4.678 | 0.794 | 0.340 | 17.75 | 1.315 | 0.398 |
| w/ **MIA**† | | **6.386** | **2.058** | **0.547** | **1.057** | **0.571** | **0.281** | **1.285** | **0.378** | 0.131 | **5.069** | 1.012 | 0.115 |
| Composers | | 37.90 | 17.40 | 5.539 | 6.916 | 3.495 | 1.584 | 14.90 | 3.974 | 0.975 | 50.02 | 11.75 | 3.426 |
| w/ MetaSGD | | 38.26 | 17.28 | 5.427 | 6.887 | 3.221 | 1.388 | 14.99 | 3.938 | 0.981 | 51.97 | 11.65 | 3.530 |
| w/ GAP | | 37.53 | 17.52 | 5.397 | 6.630 | 3.409 | 1.526 | 14.40 | 3.828 | 0.978 | 50.90 | 10.85 | 3.128 |
| w/ ALFA | | 36.53 | 14.87 | 4.115 | 5.650 | 2.770 | 1.154 | 14.18 | 3.426 | 0.799 | 42.96 | 6.814 | 1.481 |
| w/ HNPs† | | 20.38 | 9.480 | 2.413 | 2.091 | 1.075 | 0.342 | 4.497 | 0.991 | **0.111** | 19.13 | 2.698 | 0.129 |
| w/ MTNPs† | | 16.62 | 4.859 | 0.766 | 2.256 | 1.252 | 0.708 | 4.670 | 0.743 | 0.121 | 11.47 | **0.897** | **0.114** |
| w/ **MIA**† | | **5.564** | **1.844** | **0.627** | **0.975** | **0.528** | **0.237** | **1.257** | **0.343** | 0.128 | **4.715** | 0.943 | 0.156 |

$(a_n, b_n, c_n, d_n)$ shared across modalities and add per-modality Gaussian noises $\epsilon_{nm} \sim \mathcal{N}(0, 0.02)$ to control the cross-modal correlations. We use $[R^{\min}, R^{\max}] = [0.01, 0.1]$ for all modalities.

**Results.** We present the quantitative results in Table 1, where we report normalized MSEs by the scale parameter $a_{nm}$ per function, $\text{MSE} = \frac{1}{N}\sum_{n=1}^{N} \frac{1}{a_{nm}^2} \|\hat{y}_{nm} - y_{nm}\|_2^2$, following Kim et al. (2022a). The methods with no ability to handle multimodal signal jointly (CAVIA, MetaSGD, GAP, ALFA) fail to approximate the functions, showing high MSEs in all ranges of sampling ratios. In contrast, the multimodal methods (HNPs, MTNPs, and ours) are able to reconstruct each signal more precisely, even with extremely small number of support sets (*e.g.* $R < 0.02$). Moreover, while HNPs and MTNPs show strong fitting performances on smooth and low-curvature signals (*i.e.* Tanh and ReLU) when data is sufficient, our method achieves the best performances overall among the multimodal methods, verifying its effectiveness in utilizing cross-modal interactions to enhance generalization.

## 5.2 Multimodal 2D CelebA Dataset

**Setups.** We follow MTNP (Kim et al., 2022a) and conduct experiments on real-world $32 \times 32$ image function regression settings. In particular, we consider three different visual modalities of 2D facial data on CelebA (Liu et al., 2015), namely RGB images (Karras et al., 2018), surface normal maps, and sketches (Xia et al., 2021).

**Results.** Table 3 presents the quantitative results. The encoder baseline MTNPs outperforms unimodal baselines like CAVIA, MetaSGD, GAP, and ALFA in low-shot settings ($R \leq 0.25$). However, with more substantial support ($R \geq 0.50$), MTNPs fall behind ALFA, likely due to the common underfitting issues seen in encoder-based meta-learning approaches, especially with complex target functions featuring high-frequency details (Kim et al., 2019; 2022a; Guo et al., 2023; Shen et al., 2023). The poor performance of HNPs can be similarly explained, though they perform better in simpler 1D signal regressions. The original paper on HNPs (Shen et al., 2023) might not have shown this degeneracy, focusing on multimodal classification and using pretrained image features (Simonyan & Zisserman, 2014; He et al., 2016). In contrast, our MIA consistently outperforms all baselines across various support set sizes. Figure 4 demonstrates that MIA not only excels in generalization in data-scarce conditions but also retains high-frequency details effectively when data is sufficient.

## 5.3 Multimodal Climate Data

**Setups.** ERA5 (Hersbach et al., 2019) is a global climate dataset that provides hourly estimates on a wide range of atmospheric variables measured globally over a grid of equally spaced latitudes and longitudes. Out

Figure 3: Results on the multimodal 2D CelebA image function regression. We report the MSEs ($\times 10^{-3}$) computed over distinct ranges of sampling ratios. The multimodal methods are marked with the dagger (†) symbol.

| Modality | RGBs | | | | Normals | | | | Sketches | | | |
|---|---|---|---|---|---|---|---|---|---|---|---|---|
| Range $R^{\min}$ / $R^{\max}$ | 0.00 / 0.25 | 0.25 / 0.50 | 0.50 / 0.75 | 0.75 / 1.00 | 0.00 / 0.25 | 0.25 / 0.50 | 0.50 / 0.75 | 0.75 / 1.00 | 0.00 / 0.25 | 0.25 / 0.50 | 0.50 / 0.75 | 0.75 / 1.00 |
| Functa | 13.32 | 4.008 | 2.900 | 2.408 | 5.079 | 2.401 | 2.028 | 1.859 | 13.57 | 6.966 | 5.368 | 4.756 |
| w/ MetaSGD | 13.02 | 3.830 | 2.685 | 2.182 | 4.923 | 2.268 | 1.864 | 1.682 | 12.72 | 6.278 | 4.532 | 3.839 |
| w/ GAP | 12.84 | 3.726 | 2.543 | 2.024 | 4.805 | 2.184 | 1.762 | 1.570 | 12.43 | 6.023 | 4.166 | 3.407 |
| w/ ALFA | 11.83 | 3.257 | 1.957 | 1.362 | 4.115 | 1.806 | 1.285 | 1.014 | 10.80 | 4.801 | 2.463 | 1.283 |
| w/ HNPs† | 19.02 | 10.24 | 9.728 | 9.433 | 14.34 | 14.38 | 14.35 | 14.23 | 20.36 | 20.45 | 20.44 | 20.57 |
| w/ MTNPs† | 9.871 | 4.807 | 4.105 | 3.644 | 3.983 | 2.552 | 2.339 | 2.221 | 9.680 | 6.568 | 5.395 | 4.819 |
| w/ **MIA**† | **6.946** | **2.563** | **1.627** | **1.135** | **2.979** | **1.530** | **1.118** | **0.869** | **7.667** | **4.042** | **2.142** | **1.011** |
| Composers | 22.41 | 12.41 | 11.09 | 10.24 | 8.613 | 6.583 | 6.415 | 6.292 | 19.17 | 15.73 | 14.88 | 14.63 |
| w/ MetaSGD | 20.11 | 10.25 | 9.000 | 8.268 | 8.218 | 5.979 | 5.753 | 5.601 | 18.95 | 15.43 | 14.49 | 14.20 |
| w/ GAP | 20.07 | 10.05 | 8.785 | 8.039 | 8.149 | 5.847 | 5.616 | 5.461 | 18.58 | 15.24 | 14.39 | 14.15 |
| w/ ALFA | 15.12 | 4.887 | 3.376 | 2.681 | 5.444 | 2.399 | 1.773 | 1.469 | 12.57 | 5.984 | 3.416 | 2.124 |
| w/ HNPs† | 17.88 | 12.40 | 12.13 | 11.74 | 7.363 | 6.419 | 6.375 | 6.324 | 17.70 | 17.31 | 17.20 | 17.34 |
| w/ MTNPs† | 9.902 | 4.957 | 4.269 | 3.813 | 4.184 | 2.747 | 2.545 | 2.437 | 9.791 | 6.425 | 5.163 | 4.544 |
| w/ **MIA**† | **9.764** | **3.418** | **1.913** | **1.017** | **3.749** | **1.763** | **1.062** | **0.526** | **9.505** | **4.708** | **2.336** | **0.855** |

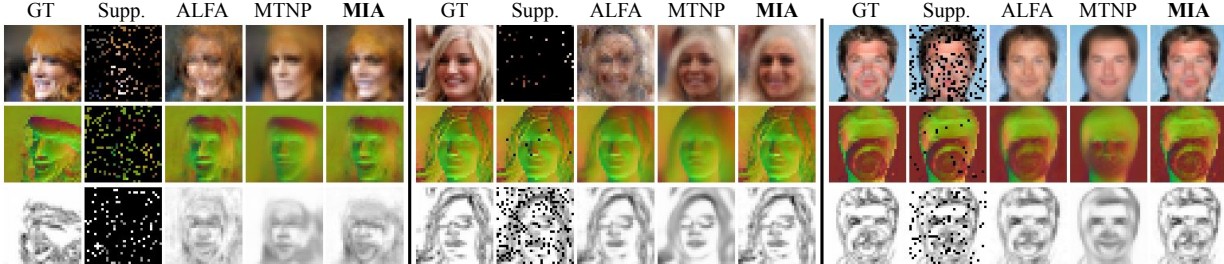

Figure 4: Three qualitative comparisons on CelebA. Compared to the baselines, MIA generalizes better in the extremely low-shot settings (1st and 2nd panes) and preserves more rich high-frequency details or nuances (2nd and 3rd panes) when sufficient data is observed.

of all the variables, we consider measurements on temperature, pressure, and humidity. Following Dupont et al. (2022c), we resize each data to $46 \times 90$ resolution.

**Results.** As indicated in Table 5, some unimodal baselines like GAP and ALFA perform well, even surpassing multimodal methods across various sampling ratios. This performance can be attributed to the relatively stable nature of atmospheric variables over time and regions (Howells & Katz, 2019), where the fast memorizing capability is beneficial. Notably, our method surpasses all baselines, affirming its capability in handling real-world multimodal climate data effectively. Supporting evidence of its superior performance is also presented in Figure 7.

### 5.4 Multimodal Audio-Visual AVMNIST Dataset

**Setups.** Unlike previous setups, modeling audiovisual signals presents unique challenges arising from the heterogeneity in coordinate systems and lack of explicit spatiotemporal alignments between modalities. We employ AVMNIST dataset (Vielzeuf et al., 2018), a collection of greyscale digit images (LeCun et al., 2010) and their pronounced audios (Jackson, 2016). We use the original images of a $28 \times 28$ resolution, while we trim the audios with a sampling rate of 2kHz and a duration of 1 second.

**Results.** In Table 6, both Functa-based CAVIA and MTNPs struggle with the audio modality, likely due to difficulties in handling high-frequency content and noise. Furthermore, the encoder baseline HNPs fails in both audio and visual signals, possibly due to inadequate fitting of complex audio without the support of expressive pretrained features, which also affects performance on simpler image functions. In contrast, our method excels in both modalities across all sampling ratios, as confirmed by Figure 7. Our approach

Figure 5: Results on ERA5 dataset in MSEs ($\times 10^{-4}$) across different sampling ratios. The multimodal methods are marked in the dagger (†) sign.

Figure 6: Results on AV-MNIST in MSEs ($\times 10^{-3}$) across different sampling ratios. The dagger (†) sign denotes multimodal methods.

| Modality | | Temperature | | | Pressure | | | Humidity | | |
|---|---|---|---|---|---|---|---|---|---|---|
| Range | $R^{\min}$ | 0.00 | 0.25 | 0.50 | 0.00 | 0.25 | 0.50 | 0.00 | 0.25 | 0.50 |
| | $R^{\max}$ | 0.25 | 0.50 | 1.00 | 0.25 | 0.50 | 1.00 | 0.25 | 0.50 | 1.00 |
| Functa | | 8.56 | 4.04 | 3.79 | 2.56 | 1.39 | 1.32 | 33.7 | 24.2 | 22.6 |
| w/ MetaSGD | | 6.89 | 3.29 | 3.10 | 1.93 | 1.01 | 0.96 | 33.1 | 21.6 | 19.5 |
| w/ GAP | | 6.30 | 2.90 | 2.69 | 1.95 | 0.98 | 0.92 | 31.2 | 18.8 | 16.0 |
| w/ ALFA | | 4.03 | 1.64 | 1.30 | 0.61 | 0.19 | 0.15 | 28.5 | 15.1 | 11.2 |
| w/ HNPs† | | 35.0 | 35.2 | 34.6 | 2.85 | 2.82 | 2.82 | 49.6 | 49.6 | 49.6 |
| w/ MTNPs† | | 4.44 | 3.65 | 3.55 | 1.42 | 1.34 | 1.32 | 28.8 | 26.0 | 25.4 |
| w/ **MIA**† | | **2.03** | **1.02** | **0.74** | **0.46** | **0.17** | **0.13** | **16.0** | **11.7** | **8.47** |
| Composers | | 8.36 | 7.58 | 7.51 | 2.70 | 2.50 | 2.50 | 46.9 | 43.2 | 42.8 |
| w/ MetaSGD | | 9.65 | 8.47 | 8.38 | 2.41 | 2.29 | 2.28 | 57.5 | 48.9 | 48.4 |
| w/ GAP | | 8.88 | 7.74 | 7.65 | 2.77 | 2.43 | 2.43 | 46.0 | 42.1 | 41.5 |
| w/ ALFA | | 7.31 | 6.05 | 6.36 | 1.95 | 1.86 | 1.83 | 38.5 | 34.3 | 34.7 |
| w/ HNPs† | | 39.6 | 40.0 | 39.3 | 4.18 | 4.16 | 4.16 | 49.4 | 49.5 | 49.5 |
| w/ MTNPs† | | 4.52 | 3.61 | 3.49 | 1.38 | 1.32 | 1.31 | 28.5 | 25.3 | 24.4 |
| w/ **MIA**† | | **3.93** | **2.48** | **1.40** | **1.13** | **0.68** | **0.41** | **24.1** | **16.6** | **7.77** |

| Modality | | Images | | | Audios | | |
|---|---|---|---|---|---|---|---|
| Range | $R^{\min}$ | 0.00 | 0.25 | 0.50 | 0.25 | 0.50 | 0.75 |
| | $R^{\max}$ | 0.25 | 0.50 | 1.00 | 0.50 | 0.75 | 1.00 |
| Functa | | 29.7 | 7.98 | 3.84 | 2.98 | 3.03 | 3.00 |
| w/ MetaSGD | | 29.8 | 7.94 | 3.71 | 0.95 | 0.50 | 0.31 |
| w/ GAP | | 29.8 | 7.97 | 3.76 | 0.93 | 0.47 | 0.29 |
| w/ ALFA | | 28.1 | 6.84 | 3.12 | 0.79 | 0.36 | 0.20 |
| w/ HNPs† | | 50.3 | 16.2 | 10.5 | 2.99 | 3.03 | 3.00 |
| w/ MTNPs† | | 28.8 | 11.4 | 6.88 | 2.37 | 2.30 | 2.23 |
| w/ **MIA**† | | **19.3** | **4.85** | **2.08** | **0.41** | **0.21** | **0.13** |
| Composers | | 33.4 | 9.86 | 2.77 | 1.02 | 0.47 | 0.22 |
| w/ MetaSGD | | 34.6 | 11.4 | 4.48 | 1.00 | 0.46 | 0.22 |
| w/ GAP | | 34.4 | 11.2 | 4.15 | 1.06 | 0.53 | 0.29 |
| w/ ALFA | | 31.9 | 8.27 | 2.46 | 1.09 | 0.48 | 0.22 |
| w/ HNPs† | | 45.6 | 18.6 | 14.8 | 2.99 | 3.03 | 3.00 |
| w/ MTNPs† | | 30.4 | 13.2 | 8.70 | 2.78 | 2.80 | 2.77 |
| w/ **MIA**† | | **18.7** | **4.24** | **1.32** | **0.65** | **0.29** | **0.15** |

Figure 7: Qualitative comparisons in ERA5 and AV-MNIST. Left: Our method achieves the lowest errors (shown in viridis color) in modeling real-world climate data. Right: Our method accurately predicts digit type with little image data and recovers the audio faithfully.

not only accurately reproduces audio signals but also effectively predicts digit classes from minimal image support, showcasing robust fitting and cross-modal generalization capabilities.

# 6 Analysis

In this section, we delve into an in-depth analysis of our method.

## 6.1 Ablation Study

**Impact of modules.** We investigate the roles of the three modules (*i.e.* USFTs, MSFTs, and Fusion MLPs) of SFTs in terms of both memorization and generalization performances. We construct ablated methods by gradually augmenting Composers with USFTs, MSFTs, and Fusion MLPs one at a time. We evaluate their fitting performances over the provided support sets and unobserved non-support parts separately, indicating the performances on memorization and generalization, respectively. We report the relative performance improvement (Maninis et al., 2019; Kim et al., 2022a) achieved by each method compared to the vanilla Composers, averaged across all signals and modalities. As in the results of Table 2a, USFTs specialize in enhancing modality-specific patterns that are useful for memorization, while MSFTs further emphasize shared information for generalization. Finally, Fusion MLPs effectively integrate the representations from USFTs and MSFTs, achieving superior performances on both memorization and generalization. Note that the reported improvement here do not specifically account for the sampling ratios of each modality, whereas the efficacy of the MSFTs becomes more apparent in cases of large imbalance in these sampling ratios across

Table 2: Ablation study on SFTs on generalization (G) and memorization (M) capability. We report relative improvement (↑) achieved by ablated methods over vanilla Composers.

| Modules | | | CelebA | | AVMNIST | |
|---|---|---|---|---|---|---|
| USFTs | MSFTs | Fusion MLPs | G | M | G | M |
| ✗ | ✗ | ✗ | 00.0 | 00.0 | 00.0 | 00.0 |
| ✓ | ✗ | ✗ | 58.3 | 92.1 | 60.7 | 84.1 |
| ✓ | ✓ | ✗ | **61.5** | 92.5 | 63.9 | 84.6 |
| ✓ | ✓ | ✓ | 61.4 | **93.0** | **65.6** | **88.7** |

(a) Impact of USFTs, MSFTs, and Fusion MLPs.

| States | | | CelebA | | AVMNIST | |
|---|---|---|---|---|---|---|
| Params | Grads | Scaling | G | M | G | M |
| ✓ | ✗ | ✗ | N/A | N/A | N/A | N/A |
| ✗ | ✓ | ✗ | 54.3 | 68.7 | 12.6 | 13.0 |
| ✗ | ✓ | ✓ | 60.6 | 91.6 | 61.7 | 87.3 |
| ✓ | ✓ | ✓ | **61.4** | **93.0** | **65.6** | **88.7** |

(b) Impact of parameters, gradients, and scaling.

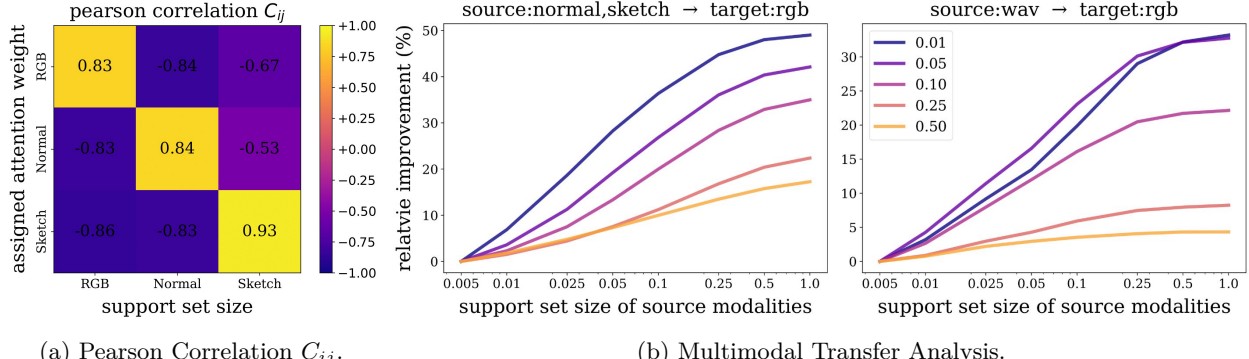

(a) Pearson Correlation $C_{ij}$.

(b) Multimodal Transfer Analysis.

Figure 8: Do our MSFTs identify and leverage cross-modal relationships as expected? Left: Pearson Correlation $C_{ij}$ between attention weights assigned to modality $i$ in MSFTs and the support set size for modality $j$. Positive correlation indicates a tendency of modality $i$ to attend more to modality $j$ when it has sufficient data. Right: Relative performance improvement (↑) when multimodal support set sizes increase. Colors indicate support set sizes of the target modality, while the x-axis represents that of the source modalities.

modalities. For a detailed analysis of this aspect, we refer the readers to discussions on Figure 8b and Table 8. Also, for comparisons and discussions on all possible ablated methods, please find Appendix E.5.

**Impact of learners' states.** We also study the impact of states utilized by SFTs by ablating parameters, gradients, or the meta-learned gradient scaling from SFTs. Similar to the previous analysis, we report their relative performance improvement compared to Composers. Table 2b reports the results. We find the parameters-only method fails entirely (reported as N/A). In contrast, the gradients-only method shows enhancements compared to vanilla Composers, and this improvement is significantly boosted with the meta-learned gradient scaling. This suggests that multimodal gradients indeed contain useful information for predicting better update directions of each INR learner, and a proper scaling technique is crucial for their effective utilization. Finally, the best performances are achieved when employing all of them. This implies that learners' current knowledge on signals (*i.e.* parameters) can be incorporated further to provide SFTs with more comprehensive view on the optimization process, leading to better parameter adjustment.

## 6.2 Adaptive cross-modal utilization of MSFTs

We validate whether our MSFTs can (1) identify valuable cross-modal patterns in learners' states and (2) utilize them to enhance the learning of INRs. For the first aspect, we examine how attention patterns of MSFTs relate to the quality of the support sets. We calculate the Pearson correlation coefficient $C_{ij}$, which reflects the relationship between attention weights assigned to the learner's state representations for modality $i$ and the support set size for modality $j$. Notably, as presented in Table 8a, we observe strong positive correlations along the diagonal. This reveals that, when observations within a specific modality suffice, MSFTs refrain from disrupting its state representations with those from other modalities. Conversely, the strong negative correlations off the diagonal imply that MSFTs tend to compensate for suboptimal state representations driven by limited data in one modality by attending more to the other modalities.

Table 3: Relative performance improvements (↑), parameter counts (↓), memory usage (↓), and meta-training time per iteration (↓) for methods considered in our paper. Notably, encoder baselines were excluded due to out-of-memory issues. The analysis illustrates increased costs with SFT integration, mitigated by higher resolutions and parameter alignment.

| Image Resolution | Method | Relative Performance Improvement (%) | # Params | Memory (GB) | Training Time (ms/it) |
|---|---|---|---|---|---|
| | CAVIA | 0.00 | 199K | 3.00 | 60.0 |
| | MetaSGD | 2.02 | 224K | 3.01 | 60.2 |
| | GAP | -2.05 | 224K | 3.01 | 66.8 |
| | ALFA | 55.1 | 299K | 3.01 | 67.2 |
| 32 × 32 | CAVIA-Large | -16.5 | 2.08M | 7.39 | 435.9 |
| | MetaSGD-Large | -10.3 | 2.08M | 7.39 | 449.2 |
| | GAP-Large | -0.69 | 2.08M | 8.00 | 425.6 |
| | ALFA-Large | 50.30 | 2.17M | 7.69 | 154.5 |
| | HNP | -4.60 | 1.90M | 58.47 | 8107.1 |
| | MTNP | 53.4 | 11.85M | 44.55 | 2031.3 |
| | **MIA** | **69.6** | 1.93M | 3.61 | 136.2 |
| | CAVIA | 0.00 | 298K | 23.69 | 765.33 |
| | ALFA | 8.86 | 644K | 24.43 | 755.90 |
| 128 × 128 | HNP | N/A | 2.01M | OOM | N/A |
| | MTNP | N/A | 21.38M | OOM | N/A |
| | **MIA** | **68.9** | 2.04M | 25.62 | 942.22 |

To examine the second aspect, we assess the performances on one target modality while the sizes of multi-modal support sets from other modalities are increased. As in previous experiments, we report the relative improvement by our method compared to the one obtained without the supports from the other modalities. Figure 8b visualizes the results that, in general, increased availability of multimodal support sets contributes to better performances in the target modality (*i.e.* increasing gains along the x-axis), while the benefit from multimodal observations becomes more significant when the size of the target support sets is smaller (*i.e.* blue lines). Based on this analysis, we confirm MSFTs' ability to correctly identify and leverage cross-modal interactions, which is particularly beneficial when modeling signals with scarce observations.

## 6.3 Computational Overhead

Our SFTs add computational complexity over vanilla CAVIA during both meta-training and meta-testing phases. To justify this, we compare the computational resources required for both phases separately.

**Meta-training Computational Overhead.** Table 3 outlines the number of parameters, memory consumption, and required meta-training time per iteration for each method, measured on the 2D multimodal CelebA dataset. Additionally, we analyze larger baseline models (CAVIA-Large, MetaSGD-Large, GAP-Large, and ALFA-Large) with parameter counts adjusted to match our MIA, along with models trained on 128 × 128 image resolutions. Unfortunately, encoder-based multimodal baselines encountered out-of-memory (OOM) issues at this larger solution setting, even with a batch size of one, and were excluded from the comparison.

As shown in the table, the integration of SFTs into our MIA framework slightly increases memory overhead and doubles the training time compared to unimodal frameworks. However, the additional resource requirements become less significant with higher resolution inputs and are offset when baseline parameters are aligned with those of MIA. This efficiency benefits from SFTs maintaining consistent computational costs relative to sample size, in contrast to the linear complexity of INRs, and a quadratic cost related to the number of INR context parameters, which scales more favorably with signal resolution as noted by Dupont et al. (2022a).

**Meta-testing Computational Overhead.** Finally, we examine the time required by the optimization-based methods (CAVIA, GAP, and ALFA) to match the 3-step adaptation performance of our method during meta-testing phase, with respect to varying support set sizes. The results are depicted in Figure 9, utilizing

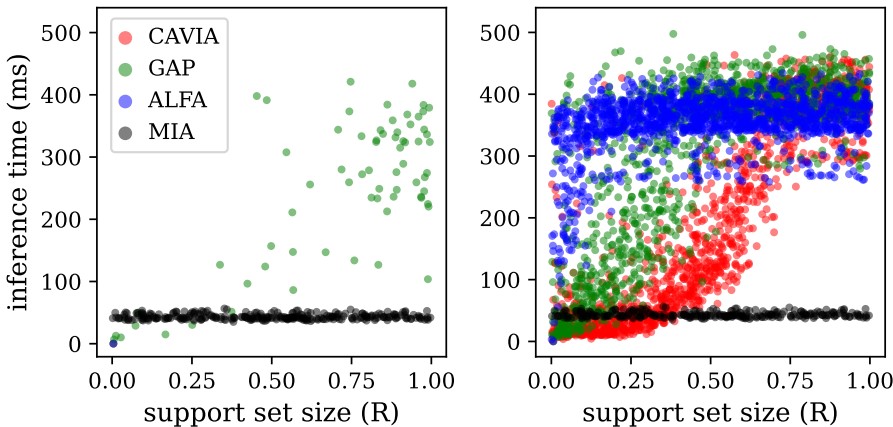

Figure 9: Inference time comparisons for baselines to achieve 3-step adaptation performances of our MIA within a 100-step optimization limit. Left: Cases where the baselines successfully match our performances. Right: For the cases where the baselines fail to match ours, we plot times for their best-achieved performances.

RGB modality in the 2D multimodal CelebA dataset. We showcase two distinct scenarios separately: one on the left where the baselines successfully match our performances, and another on the right where they fail to do so, all within the constraint of a 100-step optimization limit. In cases where the baselines do not match our performance, we report the inference time for their early-stopped best-achieved MSEs. Based on the figures, we conclude: (1) Even with 100 optimization steps, matching our 3-step performance remains a challenge for the baselines. (2) The baselines exhibit a tendency towards early overfitting with smaller support sets or underfitting with more data. (3) Even when they successfully match our performance, they often require more inference time (up to 421 ms) and optimization steps.

## 7    Discussion

In this paper, we introduce a new optimization-based meta-learning framework for INRs, substantially enhancing the performances of existing methods. This improvement is primarily attributed to our novel Multimodal Iterative Adaptation (MIA), which enables separate INR learners to share optimization processes through State Fusion Transformers (SFTs). This unique multimodal adaptation scheme leverages cross-modal interactions among learner states, facilitating rapid adaptation and boosting performance.

Despite the improvement, we also acknowledge the fundamental challenges of scaling up meta-learning algorithms. In particular, the high memory requirement necessary for bi-level optimization, such as retaining all computational graphs in the inner loop to compute exact second-order derivatives for training the meta-learners in the outer loop, hinders their application to more complex high-definition signals. To overcome such limitations, we recognize that recent advancements in the scalable meta-learning domain present viable paths to enhance the scalability of meta-learning frameworks, including ours. Notably, techniques like context pruning during meta-training (Tack et al., 2023), efficient estimation of second-order gradients (Chen et al., 2022a; Metz et al., 2022b; Choe et al., 2023; Jain et al., 2023), and the design of more efficient transformer-based meta-learners (Jain et al., 2023) have demonstrated considerable promise; through these techniques, meta-learning algorithms have been effectively extended to address more complex real-world scenarios, including the successful scaling of meta-learned INRs for $1024 \times 1024$ images or $256 \times 256 \times 32$ videos (Tack et al., 2023), and the optimization of medium to large-scale models such as BERTs (Devlin et al., 2019), RoBERTas (Liu et al., 2019), ViT-H (Dosovitskiy et al., 2021), or models even exceeding 11B+ parameters like T5-XXL (Raffel et al., 2020) (Chen et al., 2022a; Metz et al., 2022b; Choe et al., 2023; Jain et al., 2023). Additionally, exploring more efficient (Choromanski et al., 2021) or local (Liu et al., 2021) attention mechanisms, along with curriculum learning strategies such as a two-staged meta-learning scheme where we first meta-learn independent unimodal frameworks with USFTs for each modality, followed by

joint learning of the entire multimodal framework augmented with MSFTs, could be a promising avenue for reducing the computational overheads of our SFTs further. The integration of these techniques for scaling up our framework presents an intriguing direction for future research, which we intend to explore in our subsequent work.

## Acknowledgment

This work was supported by the Institute of Information & Communications Technology Planning & Evaluation (IITP) grant funded by the Korea government (MSIT) (No. RS-2021-II211343, Artificial Intelligence Graduate School Program (Seoul National University)), the IITP grant funded by the Korea government (MSIT) (No. RS-2022-II220156, Fundamental research on continual meta-learning for quality enhancement of casual videos and their 3D metaverse transformation), LG AI Research (Learning Robust and General-Purpose Multimodal Representation), and the National Research Foundation of Korea (NRF) grant funded by the Korea government (MSIT) (No. 2023R1A2C2005573).

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

## Appendix

This section provides more results and details that could not be included in the main paper.

## A More Qualitative Results

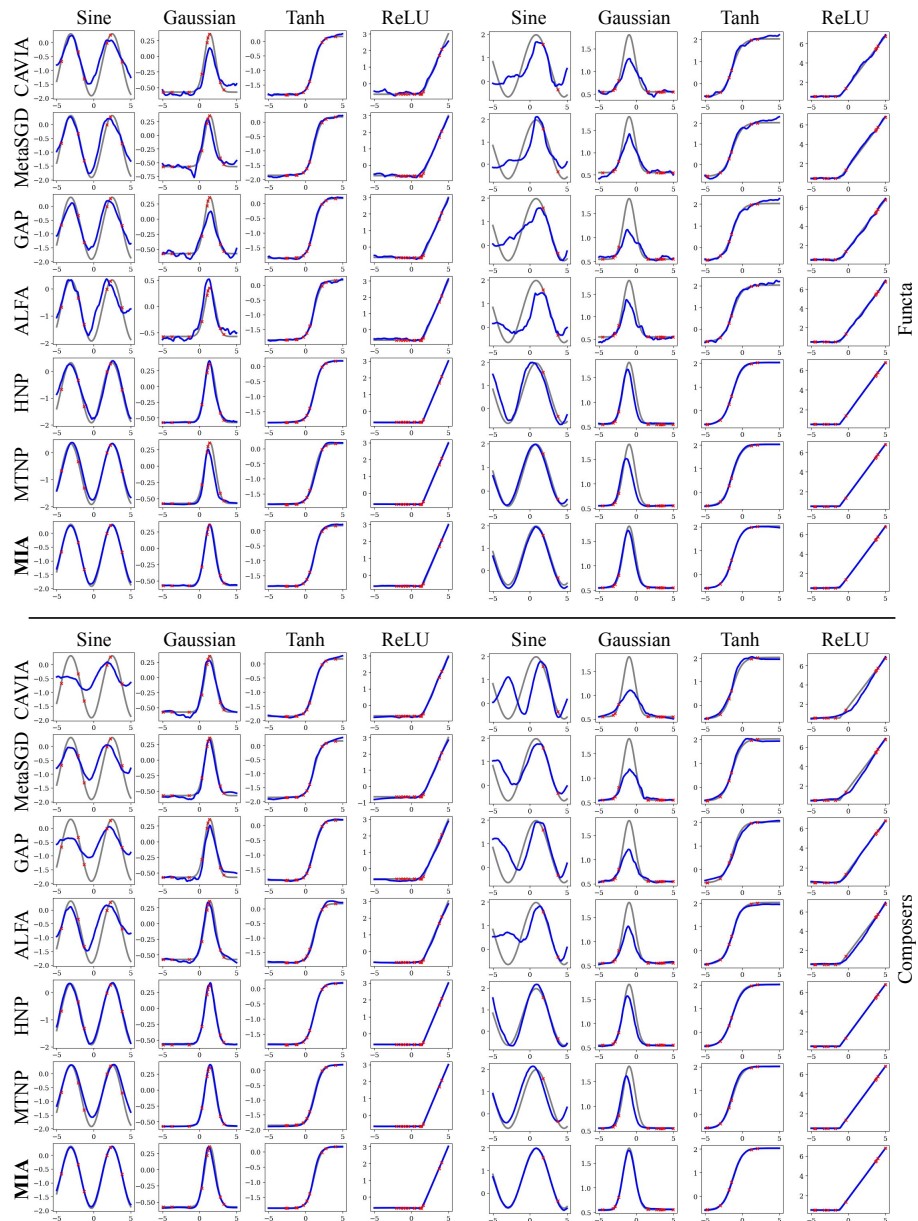

Figure 10: Qualitative comparisons on 1D synthetic functions. The black and blue lines represent the ground-truth and approximated signals recovered by each method, respectively, while red crossmarks pinpoint the locations of the provided support points. While all the methods operate well in relatively smooth and low-curvature signals (*e.g.* Tanh and ReLU), the baselines either fail entirely in Sine and Gaussian modalities (*i.e.* the unimodal approaches) or struggle to approximate them correctly (*i.e.* the multimodal baselines). Unlike the baselines, our MIA fits almost perfectly to all functions, verifying its capability in fusing multiple source of information to improve the performances.

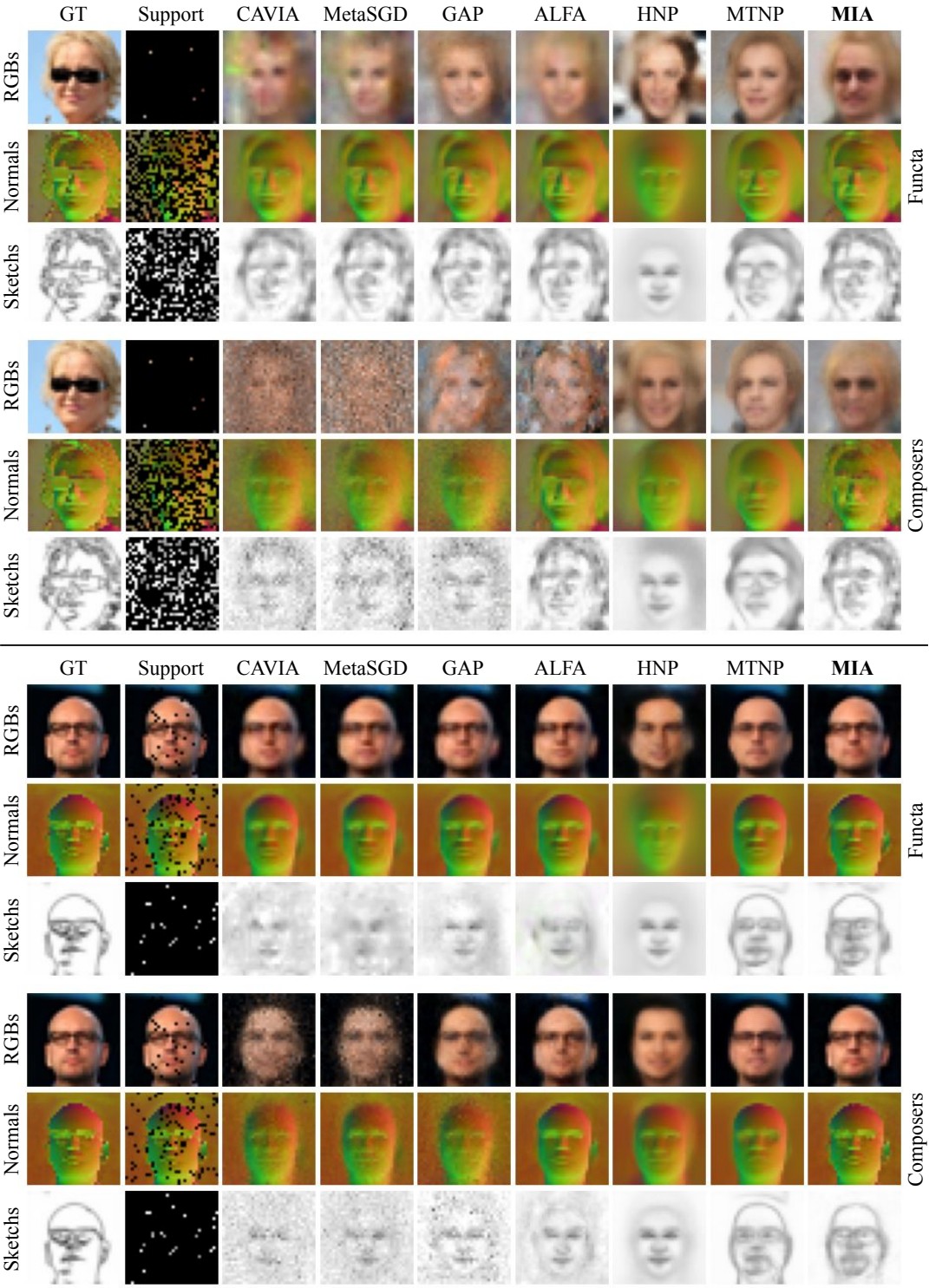

Figure 11: Qualitative comparisons on CelebA 2D visual modalities. Compared to the baselines, our MIA generalizes better in that it successfully discovers the sunglasses from the observed Normals and Sketches, followed by transferring this knowledge to RGBs for generalization (see the upper pane). Similarly, as shown in the lower pane, MIA captures the eyeglasses in RGBs and Normals and then transfer this knowledge to recover Edges more accurately.

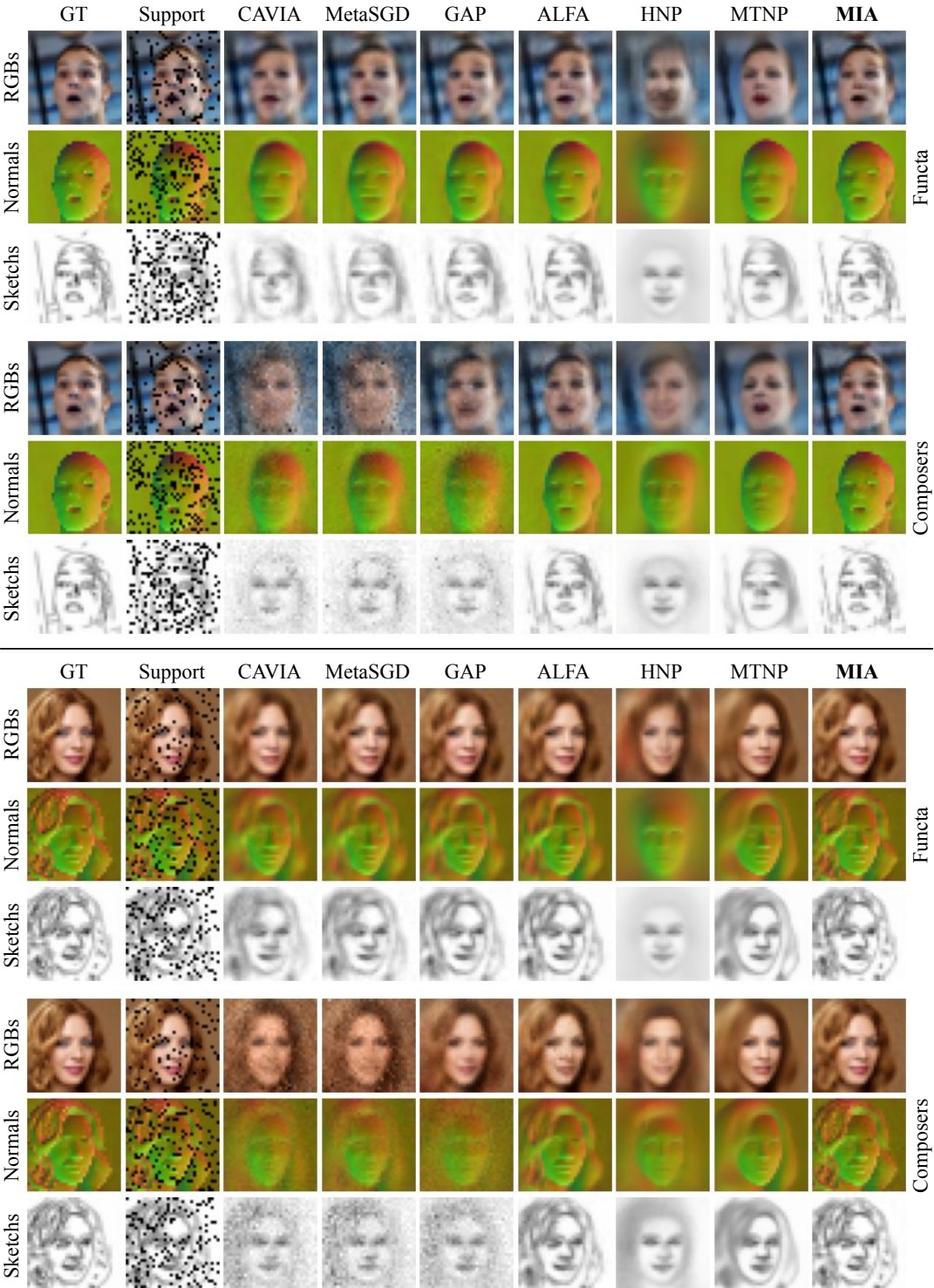

Figure 12: Qualitative comparisons on CelebA 2D visual modalities. Compared to the baselines, our MIA can capture sophisticated nuances and render high-frequency details contained in each modality signal thanks to its inheritance of iterative optimization processes from the unimodal methods. In contrast, the multimodal encoder-based methods produces overly smooth predictions or suffer from underfitting problems, due to the direct data-to-parameter mapping without optimization steps.

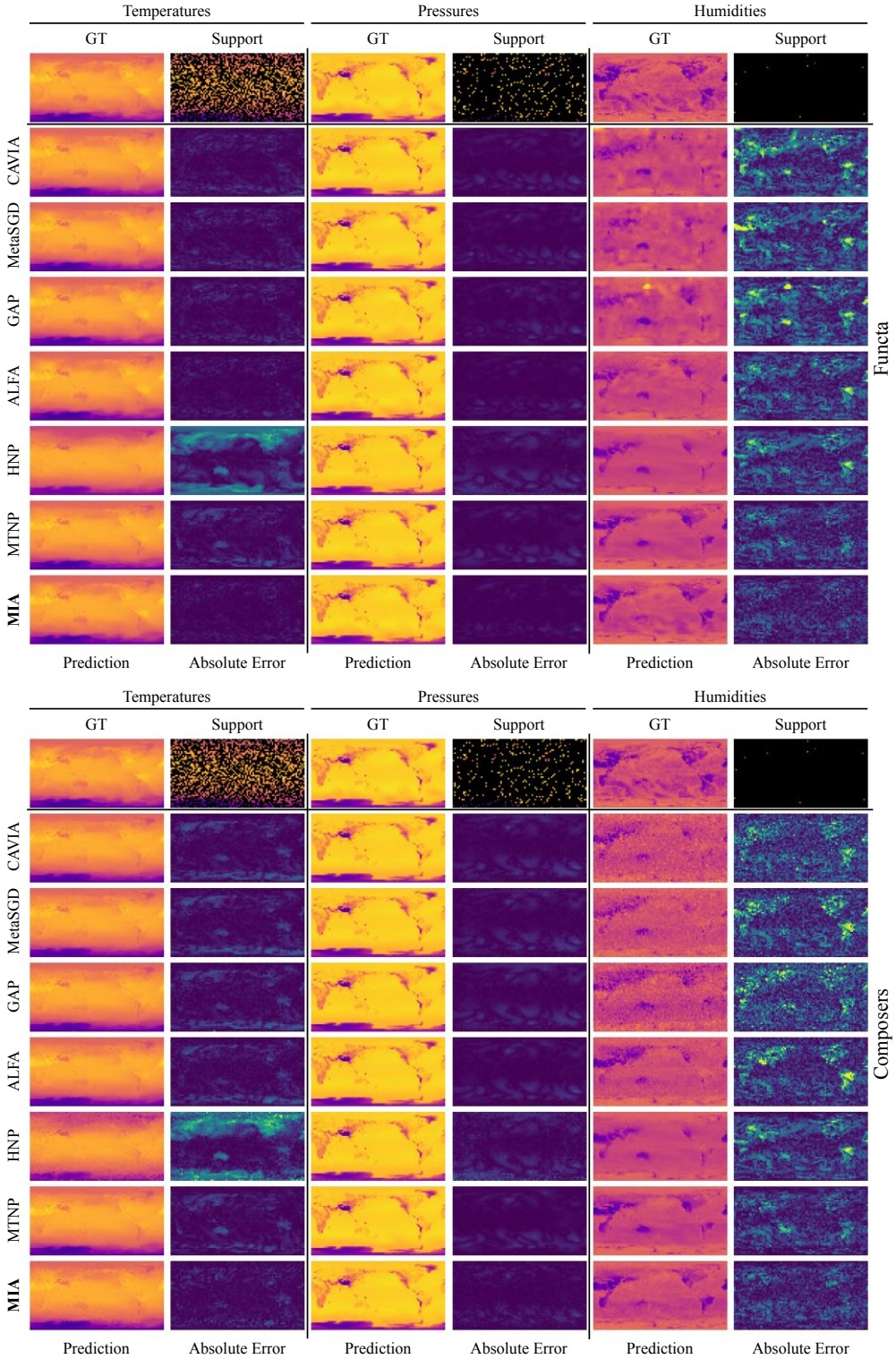

Figure 13: Qualitative comparisons on ERA5 global climate data. The first row of each pane indicates the grount-truth signals and provided support points, while the rest represents the approximations and their absolute errors achieved by each method. Our MIA shows superior generalization capability than the existing baselines, demonstrating its versatility in applications for real-world climate estimation.

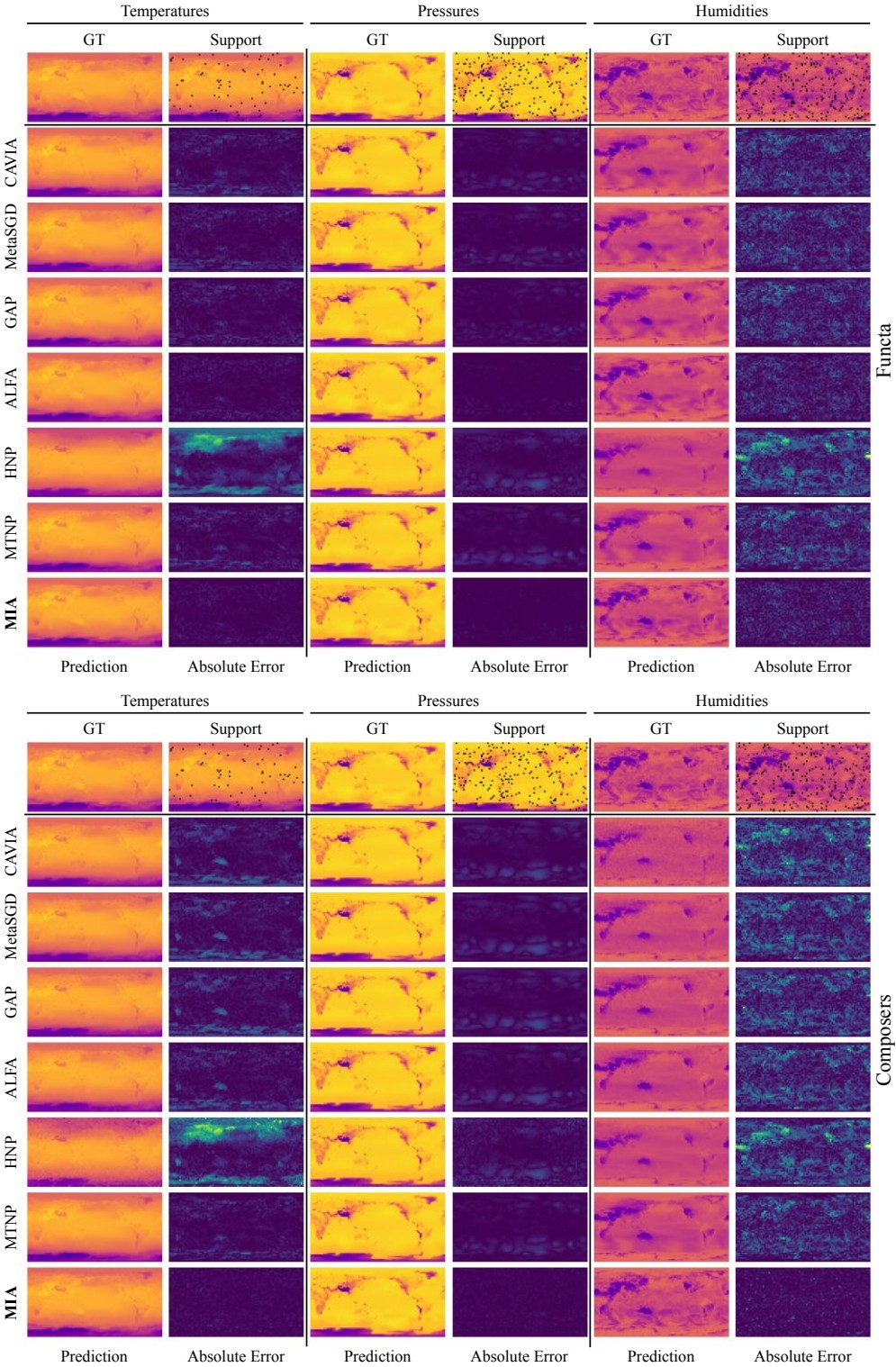

Figure 14: Qualitative comparisons on ERA5 global climate data. The first row of each pane indicates the grount-truth signals and provided support points, while the rest represents the approximations and their absolute errors achieved by each method. Our MIA shows superior convergence speed than the existing baselines, which struggle to approximate the climate data precisely and show high errors even when abundant observations are available.

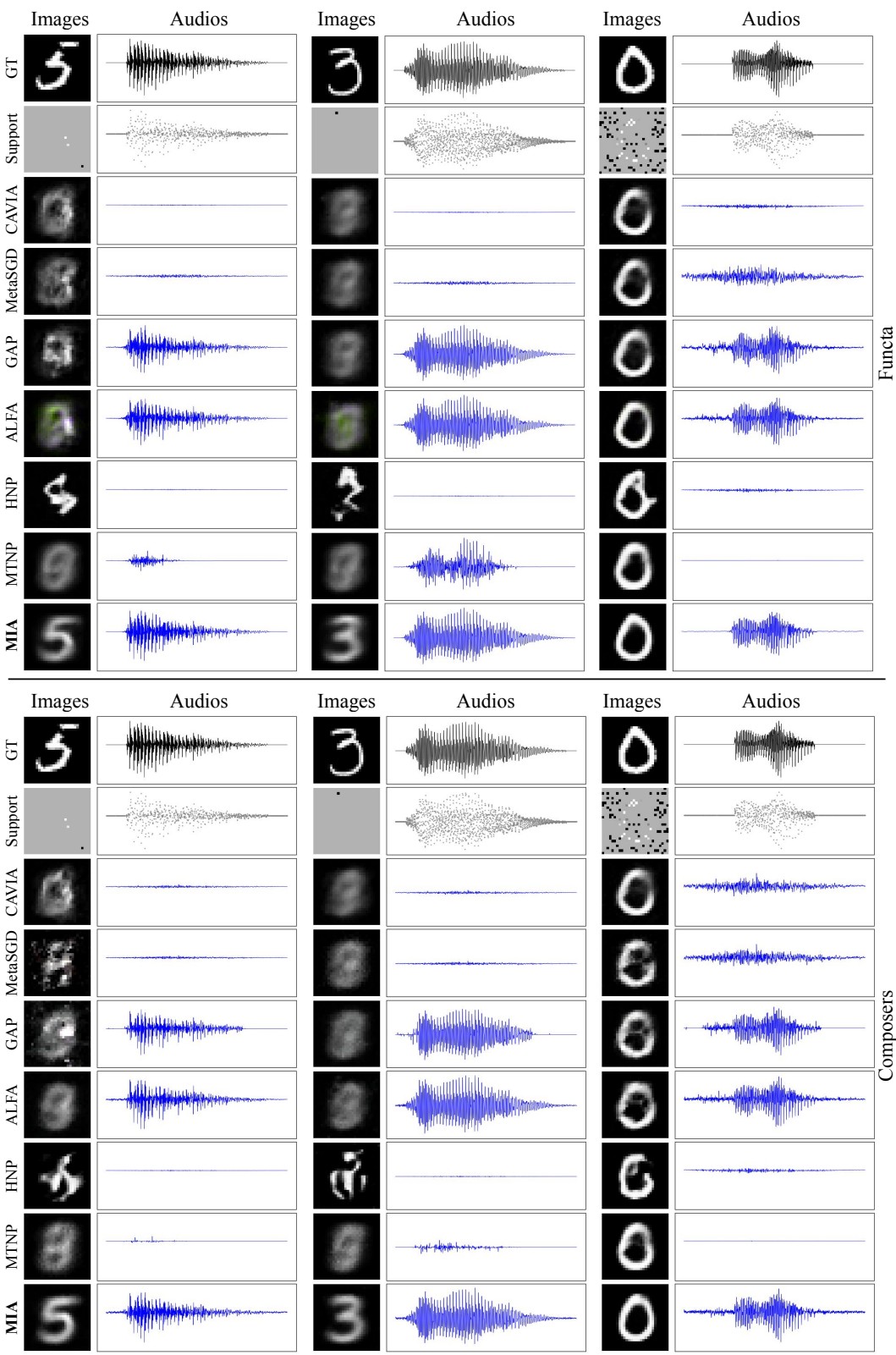

Figure 15: Qualitative comparisons on AV-MNIST. Our method successfully infer the digit classes from audio signals when little data is available from images (left two columns), while enjoying superior fitting performances when sufficient data is available (rightmost column).

# B Details of the INR frameworks

In this work, we adopt the meta-learning frameworks for INRs proposed in Functa (Dupont et al., 2022a; Bauer et al., 2023) and Composer (Kim et al., 2023). They builds upon CAVIA (Zintgraf et al., 2018), where two separate parameters are meta-learned: the INR parameters $\theta$ and the context parameters $\phi \in \mathbb{R}^{S \times D_\phi}$. In particular, the parameters $\theta$ of INRs are meta-learned to capture underlying structure of data or data-agnostic shared information across signals. In addition, the context parameters $\phi \in \mathbb{R}^{S \times D_\phi}$, a set of $D_\phi$-dimensional features, are adapted to each signal and encode per-signal variations or characteristics, which are then utilized to condition the parameters of INRs to recover the signals they are modeling. Besides the shared concepts, they differ in how they construct and utilize the context parameters to condition the INR parameters, which we detail below.

**Functa.** In Functa, the context parameters $\phi_n \in \mathbb{R}^{S \times D_\phi}$ for each signal are constructed as a spatially-arranged grid of $D_\phi$-dimensional features, where each of them encodes local variations in a signal. This grid of features is then utilized to modulate the activations of each INR layer. To do so, the set of features is first processed by an additional linear convolutional module as $\psi_n = f_{\text{conv}}(\phi_n; \theta_{\text{conv}})$, followed by bilinear/trilinear interpolation to compute a layer-wise affine transformation parameters $\psi_n(x) = \text{Interp}(x; \psi_n)$ that scale and/or shift the activations of each layer in INRs given an input coordinate $x$. In the original paper, the authors of Functa opt to adopt a shift-only modulation scheme since it shows better rate-distortion (compression vs performance) trade-offs than using both. In contrast, we adopt a scale-only modulation scheme since it is empirically shown to perform slightly better in our earlier experiments. Also, we do not modulate the activations of the first and last layers since we empirically find it stabilizes the training without hurting the performances.

**Composers.** Unlike Functa that adopt local grid of features and FiLM-like modulation scheme (Perez et al., 2018) applied to multiple layers of INRs, Composers introduce a set of non-local features $\mathbf{V}_n \in \mathbb{R}^{S \times D_\phi}$ that are used as a low-rank approximation $\mathbf{W}_n = \mathbf{U}\mathbf{V}_n$ on parameters $\mathbf{W}_n$ of a single layer in INRs. Here, $\mathbf{U} \in \mathbb{R}^{S \times D_\phi}$ is another low-rank approximation matrix that is incorporated in the INR parameters $\theta$ and captures global patterns shared across various signals, which are composed and modulated in a data-specific manner via $\mathbf{V}_n$. Following the original work (Kim et al., 2023), we approximate the second layer of INRs using $\mathbf{U}$ and $\mathbf{V}_n$.

Throughout all experiments, we use a 5-layer MLPs with 128 hidden dimensions and ReLU non-linearities in-between as a base INR architecture for both Functa and Composers. We also use random fourier features (Tancik et al., 2020) with $\sigma = 30$ to encode positional information, except that we do not use these features for the experiments on the 1D synthetic dataset.

# C Details on State Fusion Transformers (SFTs)

We construct USFTs and MSFTs using the transformer block of ViT (Dosovitskiy et al., 2021), where each of them is parameterized by one transformer block (*i.e.* $L_1 = L_2 = 1$) with a width of 192, a MLP dimension of 192, and 3 attention heads. In addition, we set the dimension of state representations $z_{nm}^{(k)} \in \mathbb{R}^{S_m \times D_z}$ to $D_z = 192$ for both USFTs and MSFTs in all experiments.

To compute the state representations $z_{nm}^{(k)}$ for each modality, we first scale the gradients by Eq (12), and then apply LayerNorm (Ba et al., 2016) to the parameters $\phi_{mn} \in \mathbb{R}^{S_m \times D_\phi}$ and the scaled gradients $g_{mn} \in \mathbb{R}^{S_m \times D_\phi}$ separately to stabilize the training. Then, we concatenate them along the feature dimension (*i.e.* $[\phi_{mn} \parallel g_{mn}] \in \mathbb{R}^{S_m \times 2D_\phi}$), followed by projecting them to the hidden space of USFTs and obtain the state representations $z_{nm}^{(k)}$ via a two-layer MLPs with a hidden dimension of 192 and ReLU activations, which is dedicated to each modality. In addition, we add positional embedding to the state representations before feeding them into USFTs, where we use sinusoidal positional encodings (Mildenhall et al., 2020) and learnable embedding for Functa and Composers, respectively. Finally, we parameterize Fusion MLPs with a 3-layer MLPs with a hidden dimension of 192 and ReLU non-linear activations. Also, we embed LayerNorm into the first and the penultimate layers of Fusion MLPs to stabilize the training.

We also attach PyTorch-style pseudo code for MIA in Listing 1 and SFTs in Listing 2, respectively.

```python
def inner_optimization_step(inr_model_dict, sft_model, modes,
                            ctx_params_dict, x_spt_dict, y_spt_dict):
    '''Code for Equations (6) and (7).
    Arguments:
        inr_model_dict: A dictionary of INRs for each modality.
        sft_model: State Fusion Transformers.
        modes: A list of modalities (e.g. ['rgb', 'normal', 'sketch']).
        ctx_params_dict: A dictionary of INRs' the context parameters for each modality.
        x_spt_dict: A dictionary of the provided support coordinates for each modality.
        y_spt_dict: A dictionary of the provided support features for each modality.
    '''
    grad_dict = dict()
    for mode in modes:
        # for each modality, obtain predicted features given support cooridnates.
        y_pred = inr_model_dict[mode](x_spt_dict[mode], ctx_params_dict[mode])
        loss = mse_loss(y_pred, y_spt_dict[mode])
        # compute gradients w.r.t context parameters.
        grad_dict[mode] = torch.autograd.grad(
            loss,
            ctx_params_dict[mode],
            create_graph = True
            # Set to `True` during meta-training, otherwise `False`.
        )[0]
    # Use SFTs to
    # 1. consider cross-modal interactions among the learners
    # 2. and enhance their weight udpates.
    # Please refer to Listing 2 for the details.
    grad_dict = fuse_states(sft_model, modes, grad_dict, ctx_params_dict)
    for mode in modes:
        # update context parameters using the enhanced gradients for each modality.
        ctx_params_dict[mode] = ctx_params_dict[mode] - grad_dict[mode]

    return ctx_params_dict
```

Listing 1: PyTorch style pseudo-code for inner optimization step via MIA.

```python
def fuse_states(sft_model, modes, grad_dict, ctx_params_dict):
    '''Code for Equations (8), (9), (10), and (12).
    Arguments:
        sft_model: State Fusion Transformers.
        modes: A list of modalities.
        grad_dict: A dictionary of the gradients w.r.t the context parameters for each modality.
        ctx_params_dict: A dictionary of the context parameters of INRs for each modality.
    '''
    ori_state_dict = dict()
    uni_state_dict = dict()
    multi_state_dict = dict()
    states = []
    for mode in modes:
        # scale gradients from each modality with the meta-learned scaling constant.
        grad_dict[mode] *= sft_model.grad_scaling[mode]
        # project gradients and context params to the hidden space
        ori_state_dict[mode] = sft_model.proj_mlp[mode](grad_dict[mode],
                                                        ctx_params_dict[mode])
        # add positional embedding and apply USFT for each modality
        state = ori_state_dict[mode] + sft_model.pos_emb[mode]
        uni_state_dict[mode] = sft_model.usft[mode](state)
        states.append(uni_state_dict[mode])

    # concatenate states for all modalities and apply MSFT
    states = torch.cat(states, dim = 1)
    states = sft_model.msft(states)
    for mode in modes:
        # calculate enhanced gradients by applying FusionMLP for each modality
        multi_state_dict[mode] = split(states, mode)
        state = torch.cat([ori_state_dict[mode],
                           uni_state_dict[mode],
                           multi_state_dict[mode]], dim = -1)
        grad_dict[mode] = sft_model.fusion_mlp[mode](state)

    return grad_dict
```

Listing 2: PyTorch style pseudo-code for State Fusion Transformers (SFTs).

Table 4: List of common configurations for each dataset.

| Hyperparameters | | Synthetic | CelebA | ERA5 | AVMNIST |
|---|---|---|---|---|---|
| modalities | | Sine, Gaussian Tanh, ReLU ($M = 4$) | RGB Normal Sketch ($M = 3$) | Temperature Pressure Humidity ($M = 3$) | Images Audios ($M = 2$) |
| number of signals ($N$) | train | 900 | 27143 | 11327 | 60000 |
| | test | 100 | 2821 | 2328 | 10000 |
| batch size | | 64 | 32 | 16 | 32 |
| epoch | | 16,000 | 300 | 300 | 300 |
| resolution | | (200, 1) | RGB - (32, 32, 3) Normal - (32, 32, 3) Sketch - (32, 32, 1) | (46, 90, 1) | Images - (28, 28, 1) Audios - (2000, 1) |
| number of original function samples ($P$) | | 200 | RGB - 1024 Normal - 1024 Sketch - 1024 | 4140 | Images - 784 Audios - 2000 |
| sampling ratio $[R^{\min}, R^{\max}]$ | | [0.01, 0.1] | [0.001, 1] | [0.001, 1] | Images - [0.001, 1] Audios - [0.250, 1] |
| outer learning rate | | $10^{-4}$ | | | |
| momentum ($\beta_1, \beta_2$) for Adam | | (0.9, 0.999) | | | |
| total inner step $K$ for optimization-based methods | | 3 | | | |
| scale for uncertainty lr | | 1 | 1 | 0.1 | 10 |
| width/depth of INRs | | 128/5 | | | |
| dimension of context parameters $\phi$ | Functa | (8, 16) | (8, 8, 16) | (8, 16, 16) | Images - (8, 8, 4) Audios - (64, 32) |
| | Composer | (8, 128) | (64, 128) | (128, 128) | Images - (64, 128) Audios - (64, 128) |
| $\sigma$ for fourier feature | | None | 30.0 | 30.0 | 30.0 |
| bootstrapping factor for evaluation | | 10 | 4 | 4 | 4 |

## D    More Experimental Details

### D.1    Common Details

In all experiments, we interpret data as a set of coordinate-feature pairs and normalize the values of both coordinates and features. Specifically, for coordinates, the values are normalized within the range $[-1, 1]^d$, where $d$ varies with the dimensionality of the dataset. For features, normalization varies by data type: values are normalized to the range $[0, 1]$ for image and climate data, and to $[-1, 1]$ for audio signals. Normalization is performed by computing the minimum and maximum values across all data points for each dataset, which are then used to scale each data point accordingly. In the case of the 1D synthetic functions, we do not normalize the features; instead, we use the normalized MSE for evaluation.

We subsample coordinate-feature pairs from the full set during the meta-training phase to promote the generalization, as well as memorization. For each signal $\mathcal{D}_{nm}$, the sampling ratio $R_{nm}$ is independently and identically drawn from a uniform distribution $\mathcal{U}(R_m^{\min}, R_m^{\max})$ for each modality $m$ to construct the support sets $\mathcal{D}_n^{\text{train}}$. For evaluation, we employ a bootstrapping technique on the meta-test dataset. We iteratively

---

**Algorithm 2** Meta-Learning Algorithm for Optimization-based Baselines.

---

**Require:** Batch size $B$; Inner step $K$; Outer learning rates $\lambda_\Theta$.

 1: Initialize $\Theta$.
 2: **repeat**
 3:     Sample a batch of $B$ joint multimodal signals $\{\{\mathcal{D}_{nm}\}_{m=1}^M\}_{n=1}^B$.
 4:     Sample $R_{nm} \sim \mathcal{U}(R_m^{\min}, R_m^{\max})$ for $\forall n, m$.
 5:     Split $\mathcal{D}_{nm}$ into $\mathcal{D}_{nm}^{\text{train}}, \mathcal{D}_{nm}^{\text{val}}$, where $|\mathcal{D}_{nm}^{\text{train}}| = P_{nm} \times R_{nm}$ and $|\mathcal{D}_{nm}^{\text{val}}| = P_{nm}$.
 6:     **for** $n = 1, \ldots, B$ **do**
 7:         **for** $k = 0, \ldots, K-1$ **do**
 8:             **for** $m = 1, \ldots, M$ **do**
 9:                 $g_{nm}^{(k)} = \nabla_{\phi_{nm}^{(k)}} \mathcal{L}(\mathcal{D}_{nm}^{\text{train}}; \theta_m, \phi_{nm}^{(k)})$
10:                 $\phi_{nm}^{(k+1)} \leftarrow \mathsf{update}(\phi_{nm}^{(k)}, g_{nm}^{(k)})$                    ▷ Inner optimization
11:             **end for**
12:         **end for**
13:     **end for**
14:     $\mathcal{L}_{\text{outer}} = \frac{1}{B} \sum_{n=1}^B \sum_{m=1}^M \mathcal{L}(\mathcal{D}_{nm}^{\text{val}}; \theta_m, \phi_{nm}^{(K)})$.
15:     $\Theta \leftarrow \Theta - \lambda_\Theta \nabla_\Theta \mathcal{L}_{\text{outer}}$.                    ▷ Outer optimization
16: **until** converged

---

sample the support and query sets multiple times for each data to mitigate potential variances that may arise from the sampling process. For 1D synthetic experiments, we apply a bootstrapping factor of 10, while for other scenarios, the factor is set to 4.

In all experiments, we use Adam optimizer (Kingma & Ba, 2014) for meta-optimization, with a learning rate of $10^{-4}$ and the momentum parameters are set as $(\beta_1, \beta_2) = (0.9, 0.999)$.

Also, we apply uncertainty-aware loss weighting technique (Kendall et al., 2018) for all multimodal methods. Please find Table 4 for the clear list of common configurations for each dataset.

### D.2    Baselines

In this section, we provide the more explanation of baselines and implementation details. To facilitate a better comparison with MIA, we include a pseudo-code for optimization-based meta-learning baselines (CAVIA, MetaSGD, GAP, ALFA) in Algorithm 2. In this context, the meta-learnable parameters $\Theta$ and the $\mathsf{update}(\cdot)$ function differ according to each baseline's unique optimization procedure.

**CAVIA.** We use a global fixed learning rate of $\alpha = 1.0$ to adapt the context parameters of CAVIA-like methods (Functa and Composers) in the inner-loop for all experiments. Here, $\Theta = \{\theta, \phi\}$, and the update rule is defined as $\mathsf{update}(\phi_{nm}^{(k)}, g_{nm}^{(k)}) = \phi_{nm}^{(k)} - \alpha_m g_{nm}^{(k)}$.

**MetaSGD.** Li et al. (2017) propose to use a meta-learned per-parameter learning rate $\alpha \in \mathbb{R}^{S \times D_\phi}$ in the inner-loop adaptation phase, which is optimized along with the meta-learner in the outer-loop optimization phase. We apply this technique to adapt the context parameters of CAVIA-based frameworks, where we initialize their initial values to 1.0 in all experiments. In this setup, $\Theta = \{\theta, \phi, \alpha\}$, and the update rule is defined as $\mathsf{update}(\phi_{nm}^{(k)}, g_{nm}^{(k)}) = \phi_{nm}^{(k)} - \alpha_m \circ g_{nm}^{(k)}$.

**GAP.** Kang et al. (2023) propose to accelerate the optimization process via Geometry-Adaptive Preconditioner (GAP), which preconditions the gradients $g^{(k)}$ at inner step $k$ by manipulating its singular values with meta-learned parameters $\mathbf{M}$. The procedure can be written as:

$$\tilde{g}^{(k)} = \mathbf{U}^{(k)}(\mathbf{M} \cdot \mathbf{\Sigma}^{(\mathbf{k})})\mathbf{V}^{(k)^{\mathrm{T}}}, \text{ where } g^{(k)} = \mathbf{U}^{(k)}\mathbf{\Sigma}^{(\mathbf{k})}\mathbf{V}^{(k)^{\mathrm{T}}}. \tag{14}$$

In the original paper (Kang et al., 2023), gradient matrix unfolding technique is introduced to facilitate SVD on the gradients of convolutional weights. Different from the original setup, now the shape of the context

parameters and their associated gradient matrices is $\mathbb{R}^{S \times D_\phi}$. Therefore, we do not using this unfolding technique and define the meta parameters as $\mathbf{M} = \text{diag}(\text{Sp}(M_1), \ldots \text{Sp}(M_{\min(S,D_\phi)}))$. In addition, we experiment with *Approximate GAP* as well and report the best performances achieved from the two methods. This approximated version bypasses the need of SVD for calculating the preconditioned gradients by $\tilde{g}^{(k)} \simeq \mathbf{M} g^{(k)}$. Here, $\Theta = \{\theta, \phi, \mathbf{M}\}$, and the update rule is defined as $\text{update}(\phi_{nm}^{(k)}, g_{nm}^{(k)}) = \phi_{nm}^{(k)} - \tilde{g}_{nm}^{(k)}$.

**ALFA.** ALFA is proposed to meta-learn the weight update procedure along with the weights to facilitate the learning. Unlike GAP, ALFA introduces an additional meta-learned neural network $h(\phi^{(k)}, g^{(k)}; \xi)$ that dynamically predicts learning rates $\alpha^{(k)}$ and weight decaying terms $\beta^{(k)}$ in a data-specific manner and for each inner step $k$, given the current parameters $\phi^{(k)}$ and their gradients $g^{(k)}$ of the learners. The resulting weight update rule can be described as follows:

$$\phi^{(k+1)} = \beta^{(k)} \cdot \phi^{(k)} - \alpha^{(k)} \cdot g^{(k)}, \quad \text{where } g^{(k)} = \nabla_{\phi^{(k)}} \mathcal{L}^{(k)}. \tag{15}$$

To construct the meta-learned network $h$, we follow the original setup (Baik et al., 2023) and parametrize it with a 2-layer MLPs with ReLU activations. Also, we reduce the context parameters and gradients by averaging them along the feature dimensions (*i.e.* $g, \phi \in \mathbb{R}^{S \times D_\phi} \to \bar{g}, \bar{\phi} \in \mathbb{R}^S$) and feed them into the meta-learned network $h$. Finally, we augment the predicted learning rates and decaying terms $\alpha^{(k)}, \beta^{(k)} \in \mathbb{R}^S$ with additional meta-learned weights $\alpha_0, \beta_0 \in \mathbb{R}^S$, as suggested in the paper. We set the initial values of $\alpha_0, \beta_0$ to 1. In summary, $\Theta = \{\theta, \phi, \xi\}$, and the update rule is defined as $\text{update}(\phi_{nm}^{(k)}, g_{nm}^{(k)}) = \beta_{nm}^{(k)} \cdot \phi_{nm}^{(k)} - \alpha_{nm}^{(k)} \cdot g_{nm}^{(k)}$, where $\alpha_{nm}^{(k)}, \beta_{nm}^{(k)} = h(\phi_{nm}^{(k)}, g_{nm}^{(k)}; \xi_m)$.

**MTNPs.** Multitask Neural Processes (MTNPs) (Kim et al., 2022a) is another class of meta-learning approach for INRs based on Neural Processes (Garnelo et al., 2018), aimed at modeling multimodal signal distributions, similar to ours. This method replaces iterative optimization steps with feed-forward encoder networks to directly predict the parameters of INRs from observed signals. For this, MTNPs adopts a dual-stream and hierarchical fusion approach to capture cross-modal relationships among signals. The first stream, driven by a latent encoder, is tasked to capture uncertainty in recovering the entire function given partial observations and to infer global latent variables shared by the observed data points for each signal in a modality ($\mathcal{D}_{mn}$). On the other hand, the second stream is guided by a deterministic encoder and is responsible for extracting local per-coordinate representations specific to each data point ($x_{nm}, y_{nm}$) that belongs to a signal in a modality. In addition, to improve the expressive power of the model, each stream is composed of a stack of specialized hierarchical multi-head attention blocks, where the earlier part captures the dependencies among data points within a modality and the latter discover their potential cross-modal relationships. For instance, in the first latent stream, cross-modal relationships among global latent variables for each modality are captured. Similarly, multimodal dependencies among local representations that belong to the same coordinate are considered in the second deterministic stream. We refer more interested readers to Section 3 and Figure 2 in the original paper of MTNPs (Kim et al., 2022a). To evaluate MTNPs in our setup, we apply two modifications to MTNPs to adapt it in our experiments: (1) We change the output dimension of the latent encoder so that it directly predicts the context parameters $\phi_{nm} \in \mathbb{R}^{S_{nm} \times D_\phi}$ that conditions the parameters of INRs. (2) For experiments on AVMNIST, we omit the module in the deterministic stream that captures cross-modal interactions among the axis-aligned local representations since there is no one-to-one correspondence between the image and audio coordinates. We use the official implementation[1] by the author to construct MTNPs.

**HNPs.** Heterogeneous Neural Processes (HNPs) (Shen et al., 2023) targets general multimodal meta-learning setups, which can be applicable to both multimodal signal regression and multimodal classification scenarios. Similar to MTNPs, HNPs has modality-specific and modality-agnostic inference modules, where the former processes per-modality observations and produces their representations, whereas the latter fuses those representations to capture cross-modal relationships. Unlike MTNPs, however, HNPs introduces additional meta-learned modality-specific and modality-agnostic latent priors, $\omega$ and $\nu$, which are respectively fed into the modality-specific and modality-agnostic inference modules as well to induce the respective latent variables, $\mathbf{z}$ and $\mathbf{w}$. Finally, the per-modality decoder takes $\mathbf{w}$ to approximate the target function. In our problem setup, we slightly modify HNPs and formulate $\mathbf{w}$ to model the parameters of INR learners $\phi$, and

---

[1]https://github.com/GitGyun/multi_task_neural_processes

use Functa and Composer for the decoder framework. We use the official implementation[2] provided by the author to construct HNPs.

## E  Additional Analysis

### E.1  Correlation between attention pattern and observation quality

Table 5: Pearson Correlation Coefficient ($C_{ij}$) between the attention scores of MSFTs assigned to the learner's state representations for modality $i$ and the support set size for modality $j$.

| $C$ | Functa | | | | Composers | | | |
|---|---|---|---|---|---|---|---|---|
| | Sine | Gaussian | Tanh | ReLU | Sine | Gaussian | Tanh | ReLU |
| Sine | 0.561 | −0.393 | −0.434 | −0.357 | 0.620 | −0.520 | −0.455 | −0.523 |
| Gaussian | −0.089 | 0.290 | −0.074 | −0.020 | −0.406 | 0.473 | −0.262 | −0.222 |
| Tanh | −0.163 | −0.068 | 0.319 | −0.036 | −0.493 | −0.439 | 0.597 | −0.421 |
| ReLU | −0.144 | −0.082 | −0.165 | 0.270 | −0.265 | −0.226 | −0.117 | 0.353 |

(a) Coefficients on multimodal 1D synthetic functions.

| $C$ | Functa | | | Composers | | |
|---|---|---|---|---|---|---|
| | RGBs | Normals | Sketchs | RGBs | Normals | Sketchs |
| RGBs | 0.779 | −0.645 | −0.579 | 0.826 | −0.837 | −0.671 |
| Normals | −0.794 | 0.843 | −0.381 | −0.826 | 0.838 | −0.529 |
| Sketchs | −0.923 | −0.891 | 0.973 | −0.856 | −0.829 | 0.933 |

(b) Coefficients on multimodal 2D CelebA images.

Figure 8a and Table 5 present when and how SFTs facilitate the interaction among the learners, quantified by Pearson Correlation Coefficient ($C_{ij}$). This coefficient measures the correlation between the attention scores of MSFTs assigned to a learner's state representations for modality $i$ and the support set size for modality $j$. Here, we describe the detailed methodology used to calculate this correlation.

Given a joint multimodal signal $\mathcal{D}_n = \{\mathcal{D}_{nm}\}_{m=1}^{M}$, $K$-step adaptation of the context parameters $\{\phi_{nm}\}_{m=1}^{M} \in R^{S \times D_\phi}$ with MSFTs of $H$ multihead attention produces the attention score map $A_n$ with a shape of $K \times H \times MS \times MS$. We reshape this attention score map to $(K \times H \times S \times S) \times M \times M$, followed by reducing the leading dimensions to obtain an average attention score matrix $\bar{A}_n$ with a shape of $M \times M$. This matrix quantifies the average interactions among the learners, where the $(i, j)$ element of this matrix amounts to the average attention scores directed from the learner of modality $i$ towards the learner of modality $j$. Finally, we compute the Pearson correlation coefficient $C$ between these average attention score maps and the sizes of the support sets $\{|\mathcal{D}_{nm}|\}_m$ across $N$ sets of joint multimodal signals, where $C_{ij}$ amounts to the correlation between $\{\bar{A}_n(i,j)\}_n$ and $\{|\mathcal{D}_{nj}|\}_n$, which is the .

### E.2  Evolving attention maps through optimization steps

We also investigate the evolving interactions among the learners through optimization steps $k = 1, \cdots, K$, as presented in Figure 17. Here, instead of computing the Pearson Correlation Coefficient between attention maps and the support set sizes as above, we quantify this degree of interactions among the learners solely by analyzing how the attention patterns within MSFTs evolve over optimization steps. This procedure involves the following steps: (1) We first reshape the attention score map $A_n$ for a signal defined in Section E.1 to $\mathbb{R}^{(H \times S \times S) \times K \times M \times M}$ (2) Then, we average this map over the first dimension to obtain $\bar{A}_n \in \mathbb{R}^{K \times M \times M}$. (3) Finally, this map is aggregated further for all signals and then averaged to $\bar{A} = \frac{1}{N} \sum_n \bar{A}_n = \{\bar{A}^k\}_{k=1}^{K}$. We visualize and analyze how this $\bar{A}^k$ evolves with $k$. Note that this final attention map contains values in the

---

[2]https://github.com/autumn9999/HNPs

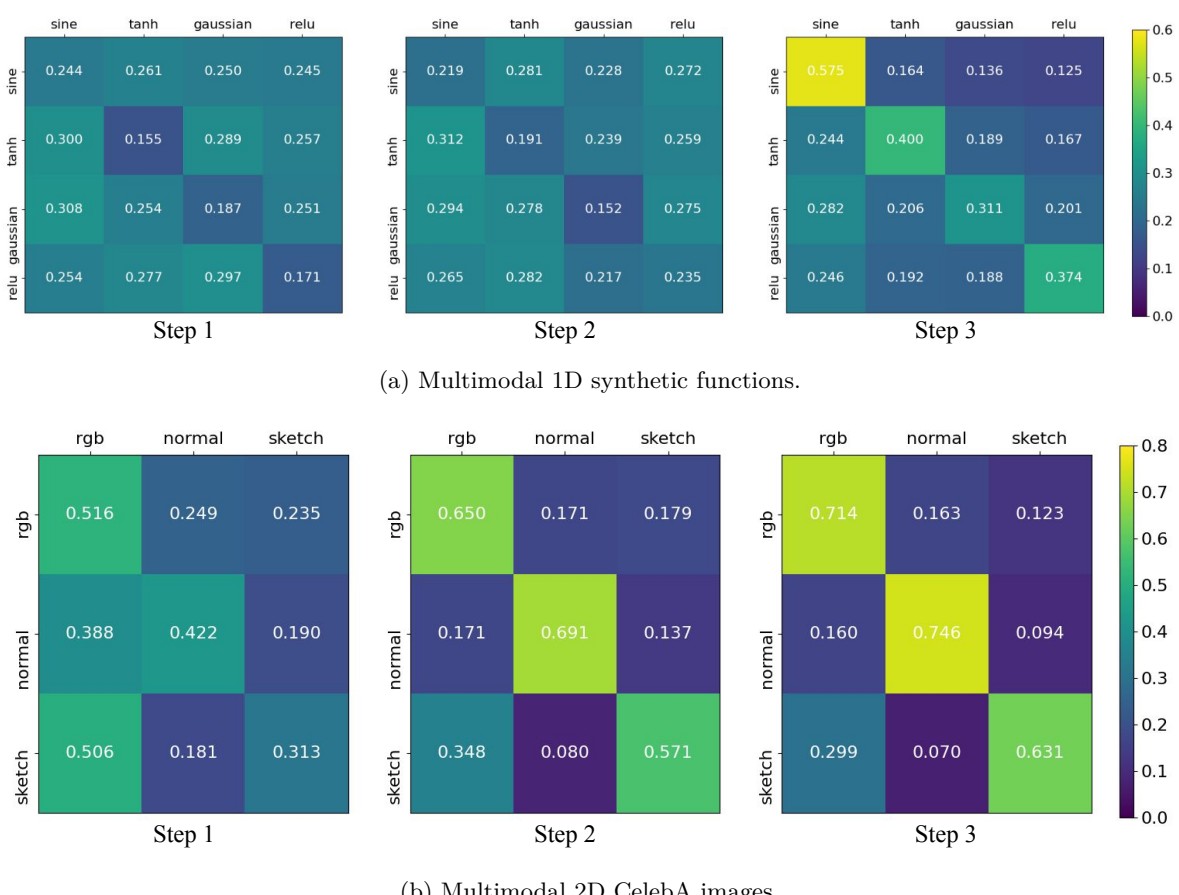

(a) Multimodal 1D synthetic functions.

(b) Multimodal 2D CelebA images.

Figure 17: Evolving interactions at each optimization step $k = 1, 2, 3$. Each $(i, j)$ element in the matrices indicates the average attention score assigned by the learner of modality $i$ to the learner of modality $j$. Please refer to Appendix E.2 for the detailed method for computing these matrices. The patterns show that learners tend to interact extensively with each other in the beginning (high off-diagonal attention scores in Step 1) and gradually attend more on themselves towards the end (high diagonal attention scores in Step 3). This highlights MIA's adaptability in ensuring a balanced utilization of multimodal interactions, emphasizing the necessity of applying multimodal adaptation iteratively for each optimization step.

range of $[0, 1]$, where higher values indicate significant cross-modal information exchange between distinct modality learners at each step, while a value of zero indicates no interaction.

The results in the figure reveal that the interactions among the learners mostly occur in the beginning of the optimization (high off-diagonal attention scores at step 1). Then, we find that the learners gradually focus more on their own states towards the end (high diagonal attention scores at step 2 and 3). This analysis shows that SFTs ensure a balanced utilization of multimodal interactions.

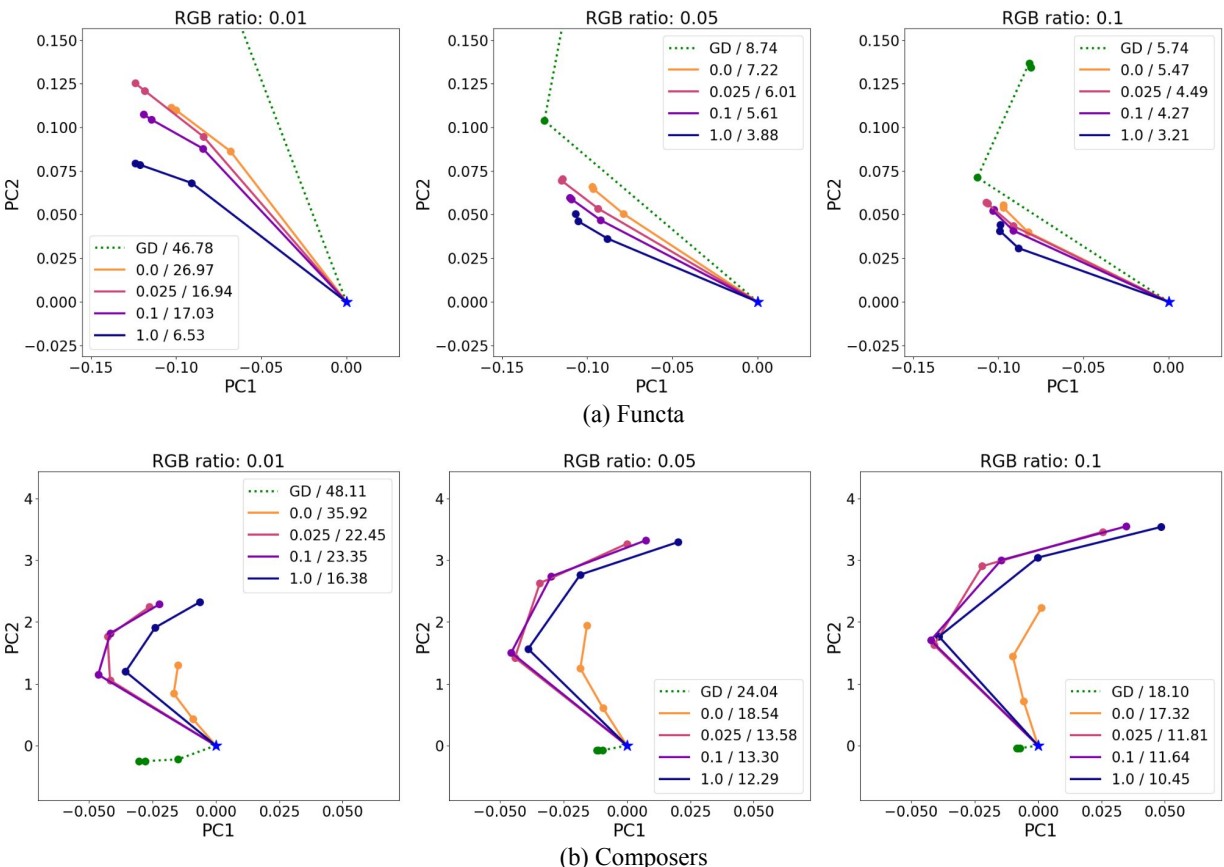

Figure 18: Learning trajectory of RGBs context parameters for the 3-step adaptation process with or without MIA, depending on the support set size of the target RGB and the sources, Normals and Sketches. For each plot, the green dotted line represents the trajectory of CAVIA adapted with the gradient descent (GD). The solid lines represent the trajectory adapted with MIA, where the different colors reflect the support set sizes of the source modalities (Normals and Sketches). The scalar values in the left side of the legend refer to the support set ratio of the source modalities and the right side means the achieved reconstruction performance (MSEs).

### E.3 Analysis study on learning trajectories via MIA

We investigate how MIA adjusts the optimization trajectories of context parameters during the adaptation process compared to CAVIA. To do this, we freeze the trained weights of CAVIA, followed by integrating it with our SFTs. Then, we meta-learn our SFTs only. This ensures that the trajectory space of the context parameters of both CAVIA and our method is aligned, thereby enabling the comparison of trajectories between the two methods. After that, we visualize the learning trajectories on both methods by applying Principal Component Analysis (PCA) on the obtained context parameters as described in Li et al. (2018).

Figure 18 illustrates that as the amount of observations from additional modalities increases, the learning trajectory with MIA shifts towards more effective solutions (lower MSE loss), moving away from bad solutions. This improvement is attributed to the SFTs' ability to enhance the sub-optimal gradients derived from limited observations by utilizing information from other modalities' learners. Conversely, when adapting through a gradient descent (GD), the learning trajectory of CAVIA remains constant regardless of observations in other modalities due to the absence of cross-modal interaction.

Table 6: Results on CelebA dataset in MSEs ($\times 10^{-3}$) and averaged unimodal attention scores ($\frac{1}{N} \sum_n \bar{A}_n(m, m)$) of MSFTs when injecting noise to gradients of other modalities' learners with varying noise level $\gamma$.

| | Functa | | | Composers | | |
|---|---|---|---|---|---|---|
| $\gamma$ | RGBs | Normals | Sketchs | RGBs | Normals | Sketchs |
| 0.00000 | 0.948 / 0.502 | 0.769 / 0.579 | 0.578 / 0.710 | 0.667 / 0.635 | 0.289 / 0.690 | 0.255 / 0.637 |
| 0.00001 | 0.976 / 0.525 | 0.774 / 0.644 | 0.581 / 0.720 | 0.664 / 0.630 | 0.288 / 0.696 | 0.255 / 0.643 |
| 0.00005 | 0.982 / 0.527 | 0.774 / 0.651 | 0.582 / 0.719 | 0.664 / 0.631 | 0.288 / 0.709 | 0.253 / 0.651 |
| 0.00010 | 0.987 / 0.527 | 0.775 / 0.653 | 0.582 / 0.718 | 0.673 / 0.656 | 0.290 / 0.727 | 0.253 / 0.658 |
| 0.00050 | 1.056 / 0.539 | 0.786 / 0.664 | 0.583 / 0.726 | 0.708 / 0.720 | 0.293 / 0.748 | 0.256 / 0.681 |
| 0.00100 | 1.092 / 0.549 | 0.796 / 0.677 | 0.584 / 0.741 | 0.714 / 0.724 | 0.293 / 0.752 | 0.256 / 0.687 |
| 0.00500 | 1.122 / 0.560 | 0.819 / 0.705 | 0.587 / 0.772 | 0.718 / 0.727 | 0.294 / 0.755 | 0.256 / 0.690 |
| 0.01000 | 1.124 / 0.561 | 0.823 / 0.709 | 0.588 / 0.775 | 0.717 / 0.727 | 0.294 / 0.755 | 0.256 / 0.690 |
| 0.10000 | 1.126 / 0.562 | 0.826 / 0.712 | 0.589 / 0.777 | 0.718 / 0.727 | 0.294 / 0.755 | 0.256 / 0.691 |
| 1.00000 | 1.127 / 0.562 | 0.826 / 0.713 | 0.589 / 0.778 | 0.718 / 0.727 | 0.294 / 0.755 | 0.256 / 0.691 |

### E.4 Analysis study on negative transfer from other modalities

In this study, we explore the impact of noisy state information from a source modality $m'$ on a target modality $m$. Our evaluation focuses on the robustness of SFTs against the potential negative transfer of incorrect information from source modalities to the target. To assess this, we inject varying levels of Gaussian noise $\epsilon_{nm'} \sim \mathcal{N}(0, \gamma I)$ into the gradient $g_{nm'}^{(k)}$ of source modalities before they are fed into SFTs. As the noise level $\gamma$ increases, the relevance of the state information from the sources to the target learner diminishes. To quantify the effect of noises, we measure the average MSEs and the average attention scores ($\frac{1}{N} \sum_n \bar{A}_n(m, m)$) within target learners' states, while varying the noise level from $10^{-5}$ to $1.0$.

As shown in Table 6, while there is a marginal rise in MSEs upon the introduction of noise, no additional performance degradation is observed beyond a certain noise level. Interestingly, as the gradient noise from learners of other modalities increases, there is a concurrent increase in unimodal attention scores for the learner of the target modality. This indicates that SFTs inherently have the ability to handle potential negative transfers across modalities.

### E.5 More in-depth ablation study for components of SFTs

Table 7: Ablation study on the components in SFTs on generalization and memorization ability. We report relative performance improvement ($\uparrow$) achieved by ablated methods (2-8) over vanilla Composers (1) on multimodal 1D synthetic function dataset.

| | Modules | | | Synthetic | |
|---|---|---|---|---|---|
| | USFTs | MSFT | FusionMLPs | Generalization | Memorization |
| (1) | ✗ | ✗ | ✗ | 0.00 | 0.00 |
| (2) | ✗ | ✗ | ✓ | 38.9 | 54.2 |
| (3) | ✓ | ✗ | ✗ | 44.4 | 63.6 |
| (4) | ✓ | ✗ | ✓ | 43.1 | 68.0 |
| (5) | ✗ | ✓ | ✗ | 86.8 | 71.6 |
| (6) | ✗ | ✓ | ✓ | 86.8 | 76.7 |
| (7) | ✓ | ✓ | ✗ | 86.7 | 75.5 |
| (8) | ✓ | ✓ | ✓ | **88.7** | **81.6** |

This section provides more in-depth ablation study results that could not be included in Table 2a. We first present the ablation study on the synthetic dataset that encompasses all possible component combinations

to investigate how each of them contribute to performance. The results are shown in Table 7. The table shows the substantial impact of FusionMLPs in enhancing the weight updates of vanilla Composers when used independently (1 vs 2) or in combination with USFTs, MSFTs, or both (3 vs 4, 5 vs 6, 7 vs 8). This demonstrates the general versatility of FusionMLPs in enhancing gradient directions or magnitudes. Moreover, we observe that incorporating USFTs significantly enhances memorization capabilities (1-2 vs 3-4). This emphasizes the advantageous role of USFTs in capturing modality-specific patterns within the learners' states. In contrast, MSFTs excel in leveraging cross-modal interactions among the learners' states, leading to a substantial improvement in the generalization performances of Composers (3-4 vs 5-8). Lastly, the most optimal performance is achieved when utilizing a combination of USFTs, MSFTs, and FusionMLPs, validating unique and indispensable roles of each component within SFTs.

Next, we extend the ablation study in Table 2a and present more in-depth analysis results on the multi-modal 2D CelebA dataset. We first compare the meta-training overheads of the ablated methods, namely the number of parameters, memory consumption, and meta-training time per iteration. The results are summarized in Table 8a. The table shows that the overheads of our full model are comparable to the other ablated methods.

Finally, we delve into the role of MSFTs and Fusion MLPs in enhancing the generalization capability of CAVIA and preventing negative transfer during the cross-modal interactions conducted within MSFTs. For this, we compare the relative performance improvement of the ablated methods for each target modality (either RGB, Normal, or Sketch) over the vanilla CAVIA while varying the sampling ratios of observable support sets from both the target and source modalities separately. The sampling ratios for the target and sources are set to either 0.01 (limited) or 0.25 (sufficient). The results are presented in Table 8b and Table 8c.

The results demonstrate that, when the support for the target modality is highly limited (*i.e.*, $R = 0.01$), CAVIA augmented with MSFTs significantly boosts the performance of their unimodal counterparts, CAVIA and CAVIA with USFTs, regardless of the sampling ratios of the source modalities. This suggests that MSFTs are indeed essential for enhancing generalization capabilities when observations are limited.

On the other hand, when observations for the target are sufficient (*i.e.*, $R = 0.25$) and those from the sources are limited (*i.e.*, $R = 0.01$), CAVIA with USFTs achieves better performance than CAVIA with MSFTs or CAVIA with both USFTs and MSFTs. This could be attributed to potential negative transfer from insufficient sources to the target. Finally, our full model alleviates this issue and achieves the best performances overall, indicating that Fusion MLPs are necessary to mitigate potential negative transfer during cross-modal interactions.

### E.6    Comparisons with other multimodal meta-learning studies

Multimodal meta-learning is not new in the domain of meta-learning. However, existing studies (Vuorio et al., 2018; 2019; Abdollahzadeh et al., 2021; Sun et al., 2023) differ significantly from our work in terms of the notion of modality and problem setups. This not only makes direct comparisons challenging but also hinders fair comparisons since evaluating their frameworks or ours to within each other's context requires substantial modifications in the methodologies. Nonetheless, in this section, we describe the problem setup and the method focused by these studies, followed by their comparative evaluation results in the context of our joint multimodal signal modeling scenarios.

**MMAML & KML**. Unlike the prevalent focus of meta-learning and meta-testing on single-domain problems, exemplified by few-shot classification tasks on a single individual dataset like Omniglot, mini-Imagenet, or CUB datasets, works such as Vuorio et al. (2018; 2019) and Abdollahzadeh et al. (2021) extend their scope to encompass multiple domains of datasets. For example, they explore simple regression problems across a union of sinusoidal, linear, and tanh functions, few-shot classification tasks combining Omniglot, mini-Imagenet, and CUB datasets, or reinforcement learning scenarios across various environments such as Point Mass, Reacher, and Ant. In these studies, each domain, whether dataset or environment, is treated as a distinct modality, leading to their concept of multimodal meta-learning. Importantly, this concept of multi-modality differs from the traditional understanding related to data types, as explicitly clarified in Section 3 of Abdollahzadeh et al. (2021).

Table 8: Comparisons on computation overheads and performances across ablated methods. Overheads are measured by parameter counts (↓), memory usage (↓), and meta-training time per iteration (↓). Performances are reported as relative performance improvements (↑).

| Modules | | | Functa | | | Composers | | |
|---|---|---|---|---|---|---|---|---|
| USFTs | MSFTs | Fusion MLPs | # Params | Memory (GB) | Training Time (ms/it) | # Params | Memory (GB) | Training Time (ms/it) |
| ✗ | ✗ | ✗ | 373K | 8.54 | 251.3 | 224K | 3.00 | 60.00 |
| ✓ | ✗ | ✗ | 1.22M | 8.76 | 266.7 | 1.26M | 3.27 | 124.8 |
| ✗ | ✓ | ✗ | 772K | 8.75 | 264.6 | 819K | 3.30 | 115.1 |
| ✓ | ✓ | ✗ | 1.44M | 8.92 | 274.7 | 1.49M | 3.48 | 133.9 |
| ✓ | ✓ | ✓ | 1.89M | 9.07 | 279.3 | 1.93M | 3.61 | 136.2 |

(a) Computational overhead

| Modules | | | RGB: 0.01 | | Normal: 0.01 | | Sketch: 0.01 | | RGB: 0.25 | | Normal: 0.25 | | Sketch: 0.25 | |
|---|---|---|---|---|---|---|---|---|---|---|---|---|---|---|
| USFTs | MSFTs | Fusion MLPs | Source: 0.01 | Source: 0.25 | Source: 0.01 | Source: 0.25 | Source: 0.01 | Source: 0.25 | Source: 0.01 | Source: 0.25 | Source: 0.01 | Source: 0.25 | Source: 0.01 | Source: 0.25 |
| ✗ | ✗ | ✗ | 0.00 | 0.00 | 0.00 | 0.00 | 0.00 | 0.00 | 0.00 | 0.00 | 0.00 | 0.00 | 0.00 | 0.00 |
| ✓ | ✗ | ✗ | 23.01 | 23.01 | 31.22 | 31.22 | 27.02 | 27.02 | 24.76 | 24.76 | 27.58 | 27.58 | 23.53 | 23.53 |
| ✗ | ✓ | ✗ | 23.24 | 50.87 | 29.13 | 44.30 | 29.38 | 50.85 | 16.68 | 29.19 | 23.32 | 28.97 | 22.21 | 32.17 |
| ✓ | ✓ | ✗ | 26.11 | 55.44 | 32.31 | 49.13 | 30.93 | 52.88 | 20.82 | 32.43 | 26.89 | 32.20 | 23.89 | 33.82 |
| ✓ | ✓ | ✓ | 29.13 | 57.98 | 32.19 | 50.11 | 30.83 | 53.13 | 22.75 | 34.74 | 27.51 | 33.36 | 23.85 | 34.25 |

(b) Ablation study based on Functa

| Modules | | | RGB: 0.01 | | Normal: 0.01 | | Sketch: 0.01 | | RGB: 0.25 | | Normal: 0.25 | | Sketch: 0.25 | |
|---|---|---|---|---|---|---|---|---|---|---|---|---|---|---|
| USFTs | MSFTs | Fusion MLPs | Source: 0.01 | Source: 0.25 | Source: 0.01 | Source: 0.25 | Source: 0.01 | Source: 0.25 | Source: 0.01 | Source: 0.25 | Source: 0.01 | Source: 0.25 | Source: 0.01 | Source: 0.25 |
| ✗ | ✗ | ✗ | 0.00 | 0.00 | 0.00 | 0.00 | 0.00 | 0.00 | 0.00 | 0.00 | 0.00 | 0.00 | 0.00 | 0.00 |
| ✓ | ✗ | ✗ | 38.65 | 38.65 | 38.33 | 38.33 | 23.76 | 23.76 | 73.35 | 73.35 | 50.29 | 50.29 | 51.13 | 51.13 |
| ✗ | ✓ | ✗ | 38.55 | 48.20 | 37.68 | 45.73 | 23.58 | 37.16 | 72.19 | 73.39 | 50.51 | 51.89 | 48.82 | 52.48 |
| ✓ | ✓ | ✗ | 40.70 | 50.47 | 38.89 | 48.73 | 26.37 | 41.05 | 72.75 | 74.05 | 52.21 | 53.78 | 50.64 | 54.64 |
| ✓ | ✓ | ✓ | 41.54 | 52.70 | 41.05 | 49.88 | 26.44 | 39.85 | 74.06 | 75.16 | 53.80 | 55.09 | 50.75 | 54.10 |

(c) Ablation study based on Composers

These works highlight that meta-learning a single initialization may be suboptimal due to the multimodal nature of the problem. To address this, they introduce an additional meta-learned module known as a task encoder. The role of this task encoder is to identify the latent modality of the observed data and predict modulation parameters. These parameters guide the learned single initialization towards modality-specific initializations. Experimental results indicate that the task encoder indeed learns to identify the modality (or dataset domain), and consequently, adapting from modality-specific initializations yields better performance than relying on a modality-agnostic single initialization.

However, it's important to note that in these papers, each data point is assumed to be sampled iid from the union of datasets. As a result, the learner adapts independently to each data point, without explicitly leveraging cross-modal relationships among modalities or incorporating mechanisms for multimodal fusion. For these reasons, we categorize these approaches as inherently unimodal meta-learning methods.

To adapt their work in our problem setup, we parameterize their task encoder with the same transformer architecture backbone as our SFTs, which is shared across all modalities, and uses it to directly predict the parameters of INRs. Then, we adapt those predicted parameters for additional $k$ optimization steps.

**AMML.** Unlike the previously mentioned studies, Sun et al. (2023) delve into a scenario where multi-modality relates specifically to data types. Their primary focus lies in addressing multimodal sentiment analysis problems, where the learner is tasked with classifying discrete sentiment scores ranging from 1 to 7 or binary sentiment classes (positive or negative). Since this classification relies on diverse sources of information represented in different modalities, encompassing texts, images, and audios, the task is referred to as multimodal inference in their work.

Table 9: Quantitative comparisons on the multimodal 1D synthetic functions. We report the normalized MSEs ($\times 10^{-2}$) computed over distinct ranges of sampling ratios, averaged over 5 random seeds.

| Modality | | Sine | | | Gaussian | | | Tanh | | | ReLU | | |
|---|---|---|---|---|---|---|---|---|---|---|---|---|---|
| Range $R^{\min}$ $R^{\max}$ | | 0.01 0.02 | 0.02 0.05 | 0.05 0.10 | 0.01 0.02 | 0.02 0.05 | 0.05 0.10 | 0.01 0.02 | 0.02 0.05 | 0.05 0.10 | 0.01 0.02 | 0.02 0.05 | 0.05 0.10 |
| Functa | | 44.26 | 16.07 | 3.319 | 18.81 | 4.388 | 0.953 | 22.61 | 3.667 | 0.586 | 65.29 | 10.79 | 2.157 |
| w/ MMAML | | 43.35 | 10.09 | 1.233 | 30.87 | 4.270 | 0.698 | 17.20 | 2.351 | 0.312 | 72.79 | 5.630 | 0.261 |
| w/ AMML | | 10.12 | 4.462 | 1.717 | 1.253 | 0.959 | 0.638 | 1.719 | 0.541 | 0.267 | 7.063 | 1.254 | 0.305 |
| w/ **MIA** | | **6.386** | **2.058** | **0.547** | **1.057** | **0.571** | **0.281** | **1.285** | **0.378** | **0.131** | **5.069** | **1.012** | **0.115** |
| Composers | | 37.40 | 16.70 | 5.284 | 5.923 | 3.149 | 1.460 | 14.81 | 4.053 | 1.029 | 48.49 | 11.98 | 3.232 |
| w/ MMAML | | 44.26 | 11.63 | 1.570 | 31.09 | 4.664 | 0.895 | 16.46 | 2.330 | 0.309 | 74.75 | 5.845 | 0.294 |
| w/ AMML | | 14.57 | 9.549 | 7.706 | 1.699 | 1.462 | 1.086 | 2.411 | 1.021 | 0.725 | 10.19 | 2.340 | 0.907 |
| w/ **MIA** | | **5.564** | **1.844** | **0.627** | **0.975** | **0.528** | **0.237** | **1.257** | **0.343** | **0.128** | **4.715** | **0.943** | **0.156** |

Table 10: Quantative comparisons on the multimodal 2D CelebA image function regression. We report MSEs ($\times 10^{-3}$) computed over distinct ranges of sampling ratios, averaged over 5 random seeds.

| Modality | | RGBs | | | | Normals | | | | Sketches | | | |
|---|---|---|---|---|---|---|---|---|---|---|---|---|---|
| Range $R^{\min}$ $R^{\max}$ | | 0.00 0.25 | 0.25 0.50 | 0.50 0.75 | 0.75 1.00 | 0.00 0.25 | 0.25 0.50 | 0.50 0.75 | 0.75 1.00 | 0.00 0.25 | 0.25 0.50 | 0.50 0.75 | 0.75 1.00 |
| Functa | | 13.44 | 4.117 | 3.052 | 2.577 | 5.067 | 2.448 | 2.092 | 1.928 | 13.29 | 7.065 | 5.704 | 5.209 |
| w/ MMAML | | 11.22 | 3.775 | 2.772 | 2.264 | 4.311 | 2.266 | 1.872 | 1.672 | 10.96 | 6.021 | 4.452 | 3.751 |
| w/ AMML | | 8.110 | 2.909 | 1.853 | 1.334 | 3.341 | 1.689 | 1.243 | 1.000 | 8.192 | 4.332 | 2.278 | 1.124 |
| w/ **MIA** | | **6.946** | **2.563** | **1.627** | **1.135** | **2.979** | **1.530** | **1.118** | **0.869** | **7.667** | **4.042** | **2.142** | **1.011** |
| Composers | | 26.40 | 15.83 | 14.30 | 13.21 | 6.979 | 4.830 | 4.630 | 4.517 | 17.99 | 14.36 | 13.27 | 12.86 |
| w/ MMAML | | 12.35 | 4.228 | 2.806 | 2.029 | 4.902 | 2.567 | 2.028 | 1.725 | 11.29 | 5.757 | 3.646 | 2.521 |
| w/ AMML | | 11.28 | 5.195 | 3.971 | 3.241 | 4.209 | 2.336 | 1.745 | 1.339 | 9.813 | 5.084 | 2.727 | 1.319 |
| w/ **MIA** | | **9.764** | **3.418** | **1.913** | **1.017** | **3.749** | **1.763** | **1.062** | **0.526** | **9.505** | **4.708** | **2.336** | **0.855** |

In their work, the authors underscore the limitations of existing methodologies, pointing out their inability to consider the heterogeneous convergence properties of each unimodal encoder network. Naively adapting these encoders using identical optimization algorithms (*e.g.* SGD with the same learning rate) results in suboptimal unimodal encoder networks. Consequently, the fusion of these suboptimal representations yields unsatisfactory outcomes for the final prediction.

To address this, Sun et al. (2023) propose to include the independent adaptation of each modality-specific encoder using the unimodal classification loss within the inner loop. Following this inner-loop phase, a multimodal fusion network integrates the extracted representations derived from these individually adapted unimodal encoders. Finally, the fused representations are utilized for predicting class labels for sentimental analysis. It's also worth mentioning that Transformer structures are employed for independent modality-specific encoder (such as the pre-trained BERT for the text encoder and Vanilla Transformers for image and acoustic encoders), whereas they are not utilized for multimodal fusion.

Since their framework is not directly applicable to our joint multimodal function regression scenarios, we investigate the effectiveness of their adaptation-in-the-inner-loop followed by fusion-in-the-outer-loop scheme by applying our method's MIA only in the final optimization step.

**Results.** Each compared method is run 5 times with different random seeds on Synthetic and CelebA datasets and their results are averaged. The results are in Table 9 and 10. From the tables, we conclude the following things: (1) MMAML greatly improves the memorization performances of CAVIA (Functa and Composer) thanks to the task encoder network, while it fails to generalize better than CAVIA due to its inherently unimodal nature of MMAML. (2) AMML further improves the generalization performances of CAVIA thanks to its multimodal-fusion-in-the-outer-loop scheme. However, its performances still fall short than our MIA, demonstrating the effectiveness of joint multimodal iterative adaptation of the learners during the adaptation stages.

