# OpenReview forum: "Meta-Learning Approach for Joint Multimodal Signals with Multimodal Iterative Adaptation"
_TMLR — Accepted by TMLR_

### Review · Reviewer_B5Ai · 2024-05-26

**Summary Of Contributions:**

This work proposes Multimodal Iterative Adaptation (MIA), a framework for learning Implicit Neural Representations (INRs) for joint multimodal signals. The two primary existing approaches for this problem are (i) encoder-based meta-learning methods that leverage multimodality using attention-based fusion, or (ii) unimodal optimization-based methods that focus on learning a good initialization to speed up convergence. However, both approaches have limitations: the former suffers from slow adaptation and the inability to capture complex signals, while the latter is unable to leverage the multimodal data for better generalization in data-scarce settings. To tackle this, MIA proposes the best of the two worlds: an optimization-based meta-learning framework with an attention-mechanism, namely, the State Fusion Transformer (SFT). The SFT aggregates learning states and allows the different modalities to interact during adaptation, and results in better parameter updates during meta-learning. The authors conduct a comprehensive experimental study on various multimodal settings, and show that MIA outperforms other approaches in terms of both generalization and memorization.

**Audience:**

Yes

**Broader Impact Concerns:**

No concern.

**Claims And Evidence:**

Yes

**Requested Changes:**

I do not have major concerns with respect to this work. Some questions to the authors:
- Have they tried experiments with more complex datasets? If not, what was the main obstacle? If yes, how did they perform?
- Can the authors comment on the fact that unimodal learning in USFT mainly accounts for the performance improvement? It seems that cross-modal interactions help, but to a much lesser degree.
- In Figure 8(a) and Table 5, the Pearson correlation coefficient is negative in the off-diagonal entries, whereas in Figure 17(b) the off-diagonal entries are positive. Does Figure 17 depict something different from the Pearson correlation?

**Strengths And Weaknesses:**

Strengths
- The paper makes a nice contribution in the multimodal INR literature. The approach is a clever combination of the two primary paradigms in multimodal INR (encoder-based and optimization-based), and it is well-motivated.
- The SFT is clearly structured into three stages, where each stage corresponds to a different functionality (unimodal learning, cross-modal learning, and state fusion).
- The experimental evaluation is quite extensive. The authors compare MIA to several baseline methods, and they try two different base INR models, namely, Functa or Composers. They also use 4 datasets, 3 of which are real. With few exceptions, MIA outperforms the other methods, with the gap being particularly pronounced in the more challenging low-sampling regime.
- The authors include an interesting ablation study that discusses key aspects of the MIA framework, e.g., the relative impact of each module of the SFT, the cross-modal utilization of SFTs, and the computational overhead of MIA. The findings in the ablation study seem to confirm that the SFT can identify useful cross-modal patterns, and make use of them to improve performance in the target modallity.

Weaknesses
- The MIA framework seems hard to scale to more complex signals, e.g., images larger than 128x128. This is  fundamental challenge of meta-learning algorithms, as the authors acknowledge, but it still significantly limits he applicability of he proposed framework in complex real settings.
- Table 2(a) suggests that the relative improvement of MIA mainly comes from the unimodal learning with USFT. The multimodal MSFT provides very small additional improvement, while MP Fusion also results in rather modest improvement. Given that this framework specifically addresses the multi-modality challenge, one may have expected that a much larger part of the relative improvement would be attributed to the cross-modal interactions, but it seems that unimodal learning can in fact primarily account for the improvement performance.

---

> ### Comment · Action_Editor_EPhy · 2024-06-27
> **Please check the authors response and start the discussion**
>
> Dear reviewer B5Ai,
>
> Thank you for reviewing the submission!
>
> This is a reminder that the authors have submitted their responses to your review comments, please take a look at the responses and see if they have addressed your concerns. Please do not hesitate to start the discussion if you have any questions/concerns.
>
> Best,
>
> AE

---

> ### Author Response · Authors · 2024-06-27
>
> We have updated our manuscript to address the concerns and provide additional clarity. Specifically, we have expanded our discussions on the computational challenges and potential solutions, and we have clarified the details of Figure 17 in Appendix E.2. Furthermore, we have included comparative analysis results above that highlight the necessity of MSFTs in Table 8 in the appendix.

---

### Review · Reviewer_NAX6 · 2024-06-08

**Summary Of Contributions:**

This paper introduces a framework called Multimodal Iterative Adaptation (MIA) to improve the learning of Implicit Neural Representations (INRs) by leveraging cross-modal knowledge exchange. Current encoder-based and unimodal optimization methods for learning INRs are insufficient for capturing the complexities of real-world multimodal signals, e.g., sketch prediction. The proposed MIA framework enhances INRs by facilitating cross-modal knowledge exchange during the optimization process, using State Fusion Transformers designed to operate in the backward pass of the learners. Extensive experiments across various multimodal datasets demonstrate that MIA significantly improves generalization and memorization performance compared to existing methods.

**Audience:**

Yes

**Broader Impact Concerns:**

I don't find any Broader Impact Concerns in the current manuscript.

**Claims And Evidence:**

Yes

**Requested Changes:**

I summarize the requested changes below (Please see the Weaknesses section above for more details.):

- Add more motivating examples and discussions for the proposed MIA approach in the introduction.
- Explore the compatibility of MIA to vision-language models.
- Add algorithm boxes for MIA and the major baseline methods.

**Strengths And Weaknesses:**

Strengths:
- The paper is well-written in general.
- Studying better methods for representation learning over multi-modal inputs is of great importance.
- The experimental results of the proposed MIA approach look promising.

Weaknesses:
- After reading the entire paper, I'm still not fully convinced of the importance of INRs in practice. Why is it an important technique that deserves more research effort? And how can it improve our machine learning pipeline (e.g., by either improving learning efficiency or generalization)? I see it more as a writing/paper organization issue. One way to improve it is to add more motivating examples and discussions in the introduction.
- Most of the multimodal experiments presented in this paper are actually still limited to visual input. I wonder if MIA can also be applied to more popular multimodal inputs like vision-language models, e.g., LLaVA [1]?
- The overall procedure of the proposed MIA approach is not quite clear. It would be great to add an algorithm box.

[1] https://arxiv.org/abs/2304.08485

---

> ### Comment · Action_Editor_EPhy · 2024-06-27
> **Please consider respond to the review comments**
>
> Dear authors,
>
> This is a reminder that the reviewer has submitted their comments. Please consider submitting your response and participate in the discussion.
>
> Thanks,
>
> AE

---

> ### Author Response · Authors · 2024-06-27
>
> > Q: Add more motivating examples and discussions for the proposed MIA approach in the introduction.
>
> A: We appreciate the helpful feedback. Many real-world signals or data are inherently continuous. For instance, videos are recordings of dynamic scenes that change continuously over space and time. However, during the sampling and discretization processes in recording devices, these signals are often stored as discrete arrays or tensors. INRs are designed to approximate and recover the inherent continuity of these signals by representing them as continuous functions. Once learned, these representations are particularly useful in various practical applications requiring precise signal modeling, such as reconstruction, compression, super-resolution, and novel-view synthesis. Since many signals can be represented as continuous functions, INRs have demonstrated promise across a wide range of modalities, including audio, time-series, image, video, and 3D data. We have incorporated this discussion into the beginning of the introduction section accordingly.
>
> > Q: Explore the compatibility of MIA to vision-language models.
>
> A: We appreciate the constructive comment. As discussed above, INRs focus on the precise recovery of inherently continuous signals by adapting to their traditionally sparse and discrete counterparts, such as discrete arrays. Our framework, MIA, is specifically designed to enhance the learning of such INRs for correlated multimodal signals by facilitating cross-modal interactions across INR learners for different modalities during iterative optimization. This led us to validate the efficacy of MIA in recovering multimodal signals, including 2D visual, climate, and audio-visual data, under varying levels of observability. Unlike INRs, vision-language models like LLaVA primarily focus on tasks requiring high-level understanding or reasoning about visio-lingual input data rather than their precise reconstruction. Due to this distinction, the application of MIA to such vision-language models is less natural or beyond the scope of this work. Nevertheless, we believe exploring potential avenues for integrating MIA into collaborative scenarios with vision-language models is valuable, and we leave it for future research.
>
> > Q: Add algorithm boxes for MIA and the major baseline methods.
>
> A: We appreciate the helpful advice. While we originally provided PyTorch-style pseudo-codes for the core part of our MIA framework in the appendix, we acknowledge that their accessibility may have been limited. Consequently, we have included the entire meta-learning algorithm for our MIA framework in Algorithm 1 of the main paper, along with the algorithms for the optimization-based baselines in Appendix D.2. We hope this clarifies the meta-learning process of our MIA and the major baselines.

---

### Review · Reviewer_P66A · 2024-06-19

**Summary Of Contributions:**

The paper introduces Multimodal Iterative Adaptation (MIA), a framework that advances the learning of Implicit Neural Representations (INRs) by integrating multimodal fusion with optimization-based meta-learning. MIA addresses the limitations of encoder-based methods by enabling cross-modal knowledge exchange during iterative optimization, which enhances the generalization of INRs to complex signals. The framework’s centerpiece, State Fusion Transformers (SFTs), is an attention-based meta-learner that operates during the backward pass, aggregating learning states and predicting improved parameter updates. This approach allows for a more nuanced adaptation to multimodal signals and overcomes data scarcity challenges.

**Audience:**

Yes

**Broader Impact Concerns:**

No broader impact is discussed.

**Claims And Evidence:**

Yes

**Requested Changes:**

Please kindly refer to the Strengths And Weaknesses.

**Strengths And Weaknesses:**

Strengths:
* MIA is claimed to be the first to explore the integration of multimodal structures within the learning of multiple correlated INRs, offering a new perspective on optimizing across modalities.
* By allowing modalities to inform each other, MIA significantly improves generalization capabilities, even with limited data, outperforming unimodal and encoder-based baselines.
* The paper presents a thorough evaluation of MIA across various real-world multimodal signal regression scenarios, demonstrating its superior performance in both generalization and memorization.


Weaknesses:

* Although the paper is generally well-written, it is better for the authors to provide an application example of the INR setting where the data is scarce to make the audiences better understand the targeted tasks, especially for the ones that are not familiar with the INR research.

* While the setting focuses on the optimization of INRs when the observations are sparse, it is unclear how sparse the four datasets used in experiments are. It would be better for the authors to provide some quantitative measurements regarding to $P$ of each dataset to further demonstrate their setting matches their motivation. Meanwhile, how many INR learners used (the $n$ and $m$) should also be specified for each dataset.

* According to the results in Table 2 for the ablation study regarding the impact of modules, it seems like the USFTs is the most important module in SFTs as adding the other two modules only exhibits relative marginal improvements over USFTs-only variants. It would be beneficial for the authors to elaborate more on the rationale behind the design of both MSFTs and Fusion MLPs, especially considering together with the additional computational complexity brought by adding MSFTs and Fusion MLPs. This would be interesting for practitioners in employing the proposed method in industrial applications.

* One minor suggestion is, given the designs of the MIA and SFTs are relatively intuitive, it would be interesting to see some theoretical groundings behind the designs with the current motivation.

---

> ### Comment · Action_Editor_EPhy · 2024-06-27
> **Please consider respond to the review comments**
>
> Dear authors,
>
> This is a reminder that the reviewer has submitted their comments. Please consider submitting your response and participate in the discussion.
>
> Thanks,
>
> AE

---

> > ### Author Response · Authors · 2024-06-27
> >
> > > Q: According to the results in Table 2 for the ablation study regarding the impact of modules, it seems like the USFTs is the most important module in SFTs as adding the other two modules only exhibits relative marginal improvements over USFTs-only variants. It would be beneficial for the authors to elaborate more on the rationale behind the design of both MSFTs and Fusion MLPs, especially considering together with the additional computational complexity brought by adding MSFTs and Fusion MLPs. This would be interesting for practitioners in employing the proposed method in industrial applications.
> >
> > A: We acknowledge that the rationale behind our MSFTs and FusionMLPs may not have been clearly demonstrated in our original presentation. Their efficacy becomes increasingly evident in scenarios with large imbalances in sampling ratios across modalities. As analyzed in Section 6.2, MSFTs enhance the generalization capability of the INRs being learned by incorporating complementary information from other modalities when observations are sparse in one modality. This analysis further validates their robustness to negative transfer from source modalities, ensuring that integrating additional information from source modalities does not adversely affect the performance of target modalities.
> >
> > The impact of MSFTs and FusionMLPs is not fully captured in Table 2 for two primary reasons: (1) In our meta-test sets for the multimodal 2D CelebA dataset, cases with such imbalances represent a relatively low portion, about 13% of the total dataset. (2) As a result, the overall average performance improvement appears marginal due to the dilution from cases that do not significantly benefit from MSFTs. Despite these limitations, we opt to report average performance improvement across all signals and modalities to provide a comprehensive overview of the model's behavior across all test scenarios.
> >
> > To address this limitation, we updated our manuscript to incorporate the analysis results specifically targeting scenarios with large imbalances in order to better elucidate the role of MSFTs and Fusion MLPs, along with their added computational overheads. Please find discussion in Appendix E.5 and Table 8 in the appendix. The results clearly demonstrate that MSFTs are crucial for enhancing generalization capabilities and that FusionMLPs are necessary to mitigate potential negative transfer during cross-modal interactions.
> >
> >
> > > Q: One minor suggestion is, given the designs of the MIA and SFTs are relatively intuitive, it would be interesting to see some theoretical groundings behind the designs with the current motivation.
> >
> > A: The motivation behind our MIA and the design choices for SFTs are grounded in the empirical success observed in the domain of Learning-to-Optimize (L2O). L2O [1] aims to meta-learn optimizers that enhance the learning of neural networks, moving beyond conventional handcrafted optimizers like SGD and Adam. Recent works in L2O [2,3] further suggest that transformers can serve as such learned optimizers in this context. Unfortunately, neither learned nor handcrafted optimizers currently offer theoretical guarantees of convergence and optimality within the scope of non-convex optimization for deep learning. That said, we acknowledge the value of such theoretically guaranteed frameworks. We hope our work will inspire future research to address these gaps and further refine the practical utility and theoretical foundations of L2O methodologies.
> >
> > References:
> > [1] Andrychowicz et al., Learning to learn by gradient descent by gradient descent, in NeurIPS, 2016.
> > [2] Gärtner et al., Transformer-based Learned Optimization, In CVPR, 2023.
> > [3] Moudgil et al., Learning to Optimize with Recurrent Hierarchical Transformers, In ICML Workshop, 2023.

---

> ### Author Response · Authors · 2024-06-27
>
> > Q: Although the paper is generally well-written, it is better for the authors to provide an application example of the INR setting where the data is scarce to make the audiences better understand the targeted tasks, especially for the ones that are not familiar with the INR research.
>
> A: We appreciate the thoughtful suggestion. The major focus of INRs is on the precise recovery of inherently continuous signals from their traditionally sparse and discrete counterparts. For example, while most real-world videos are recordings of dynamic scenes that change smoothly in space and time, they are sampled sparsely and stored as a sequence of discrete 2D image arrays via recording devices. With that being said, the task of learning INRs is inherently aimed at reconstructing these continuous signals from limited discrete samples, which we often refer to as data. We included such an explanation in the beginning of the introduction section accordingly.
>
> > Q: While the setting focuses on the optimization of INRs when the observations are sparse, it is unclear how sparse the four datasets used in experiments are. It would be better for the authors to provide some quantitative measurements regarding to $P$ of each dataset to further demonstrate their setting matches their motivation. Meanwhile, how many INR learners used (the $n$ and $m$) should also be specified for each dataset.
>
> A: We appreciate the constructive suggestion. We revised our manuscript and specified the number of modalities $M$, signals $N$, and their coordinate-feature pairs $P_{nm}$ in Table 4 in the appendix.
>
> Let us clarify the sparsity more. In our setup, each dataset consists of $N$ joint multimodal signals with $M$ different modalities. For the $n$-th signal from the $m$-th modality, there are $P_{nm}$ coordinate-feature pairs. During meta-training and meta-testing, we ensured the observable support sets exhibited varying degrees of sparsity by subsampling coordinate-feature pairs using a sampling ratio $R_{nm} \in [0, 1]$ for each signal independently, where a lower $R_{nm}$ indicates higher sparsity. This resulted in support sets containing $P_{nm} \times R_{nm}$ coordinate-feature pairs.
>
> Once INRs were learned using these support sets, they were evaluated based on the query sets, which are essentially the original signals with $P_{nm}$ coordinate-feature pairs. In our paper, when comparing the models’ meta-testing performances under varying levels of sparsity, we consistently used the sampling ratio $R_{nm}$ to indicate the sparsity of the observable support sets used for adaptation, as both the total number of coordinate-feature pairs $P_{nm}$ and the actual sizes of the support sets may vary across datasets and modalities.
>
> We hope this clarifies that our experimental setup matches our motivation.

---

### Comment · Action_Editor_EPhy · 2024-07-04
**Please check the authors response and start the discussion**

Dear **reviewers**,

Thank you for reviewing the submission.

Now the authors have submitted their responses to your review comments, please take a look at the responses at your earliest convenience, and see if they have addressed your concerns.

Please do not hesitate to start the discussion if you have any questions/concerns.

Best,

AE

---

### Decision · Action_Editor_EPhy · 2024-07-27

**Recommendation:** Accept as is

**Comment:**

In this paper, the authors presented a new meta-learning framework for multimodal modelling. Specifically, a Multimodal Iterative Adaptation (MIA) approach was proposed that combined multimodal fusion with optimization-based meta-learning. Extensive experimental evaluations on several real-world scenarios show the effectiveness of the proposed method. The paper is generally well-written and easy to follow.

Three expert reviewers were invited to review the paper, and both strengths and weaknesses were raised. With back-and-forth discussions and revision, most of the concerns were well addressed by the authors in the revised version, and all the reviewers are satisfied with the revision. In the end, a positive score (with two *Leaning Accept* and one *Accept*) was recommended.
On the other hand, both the reviewers and AE found that the interesting approach and the contributions made in this paper could be of interest to a group of audiences in TMLR. As a result, the AE is pleased to inform that the paper has been accepted to be published in TMLR.

**Audience:**

Yes, there would be a group of audience in TMLR interested in knowing the findings of this paper. Specifically, the interesting multimodal approach together with the extensive experimental analysis could be of interest to the multimodal researchers.

**Claims And Evidence:**

The claims made in the submission are supported by accurate, convincing and clear evidence in general.